# Current In Vivo Models of Varicella-Zoster Virus Neurotropism

**DOI:** 10.3390/v11060502

**Published:** 2019-05-31

**Authors:** Ravi Mahalingam, Anne Gershon, Michael Gershon, Jeffrey I. Cohen, Ann Arvin, Leigh Zerboni, Hua Zhu, Wayne Gray, Ilhem Messaoudi, Vicki Traina-Dorge

**Affiliations:** 1Department of Neurology, University of Colorado School of Medicine, Anschutz Medical Campus, Aurora, CO 80045, USA; 2Department of Pediatrics, Columbia University College of Physicians and Surgeons, New York, NY 10032, USA; aag1@cumc.columbia.edu; 3Department of Microbiology, Columbia University College of Physicians and Surgeons, New York, NY 10032, USA; mdg4@cumc.columbia.edu; 4Laboratory of Infectious Diseases, National Institute of Allergy and Infectious Diseases, National Institutes of Health, Bethesda, MD 20892, USA; jcohen@niaid.nih.gov; 5Pediatric Infectious Disease, Stanford University, Stanford, CA 94305, USA; aarvin@stanford.edu (A.A.); zerboni@stanford.edu (L.Z.); 6Department of Microbiology, Biochemistry and Molecular Genetics, New Jersey Medical School, Rutgers University, Newark, NJ 07102, USA; zhuhu@njms.rutgers.edu; 7Biology Department, University of Mississippi, Oxford, MS 38677, USA; wlgray@olemiss.edu; 8Departments of Molecular Biology and Biochemistry, School of Biological Sciences, University of California Irvine, Irvine, CA 92697, USA; imessaou@uci.edu; 9Division of Microbiology, Tulane University, Tulane National Primate Research Center, Covington, LA 70433, USA; vtraina@tulane.edu

**Keywords:** varicella zoster virus, animal models, simian varicella virus

## Abstract

Varicella-zoster virus (VZV), an exclusively human herpesvirus, causes chickenpox and establishes a latent infection in ganglia, reactivating decades later to produce zoster and associated neurological complications. An understanding of VZV neurotropism in humans has long been hampered by the lack of an adequate animal model. For example, experimental inoculation of VZV in small animals including guinea pigs and cotton rats results in the infection of ganglia but not a rash. The severe combined immune deficient human (SCID-hu) model allows the study of VZV neurotropism for human neural sub-populations. Simian varicella virus (SVV) infection of rhesus macaques (RM) closely resembles both human primary VZV infection and reactivation, with analyses at early times after infection providing valuable information about the extent of viral replication and the host immune responses. Indeed, a critical role for CD4 T-cell immunity during acute SVV infection as well as reactivation has emerged based on studies using RM. Herein we discuss the results of efforts from different groups to establish an animal model of VZV neurotropism.

## 1. Introduction

Varicella-zoster virus (VZV), a member of the human alphaherpesvirus family, causes chickenpox (varicella) mostly in children and establishes a latent infection in cranial, dorsal root and autonomic ganglia. Latent VZV can reactivate decades later to produce shingles (zoster). VZV-specific cell-mediated immunity (CMI) declines with age, resulting in zoster and associated neurological complications that are also seen in immunocompromised organ transplant recipients and in patients with cancer or AIDS. The actual mechanism of reduction in VZV-specific CMI and associated virus reactivation remains unclear. VZV produces chickenpox and shingles only in humans, underscoring the need for an animal model to study VZV neuropathogenesis. Attempts by multiple groups to establish VZV infection in guinea pigs and mice by experimental inoculation have resulted in seroconversion without clinical symptoms [1,2,3,4,5,6]. On the other hand, simian varicella virus (SVV) infection in non-human primates (NHP) results in varicella. Although the NHP model is expensive, it has provided critical insights into varicella pathogenesis and neurotropism not available with any of the other models. Herein, we review the existing animal models of VZV neurotropism.

## 2. Infection of Guinea Pigs with Varicella-Zoster Virus

### 2.1. Initial Approaches

There is a strong belief, reflected in the introductions to many papers, that VZV grows only in human cells. Although humans are its only natural host and VZV was first cultured in cells of human origin [7], early studies revealed that VZV could also be propagated in cells from guinea pig embryos [8,9]; nevertheless, because VZV multiplies to a greater extent in human than in guinea pig cell cultures, human cells are almost always selected for research on VZV. Takahashi and colleagues, however, took advantage of the ability of VZV to infect guinea pig cells and passaged VZV several times in guinea pig fibroblasts as part of their effort to attenuate the virus for use as a live varicella vaccine. The first study describing the development and use of the live attenuated varicella vaccine appeared in 1974 [10] and the critical prospective study that showed its efficacy in children with leukemia who were at risk of death from varicella appeared in 1984 [11]. This vaccine, now given in two doses [12], has stably retained its efficacy [13] and has been modified to protect at-risk elderly individuals from zoster [14].

### 2.2. First Guinea Pig Models of VZV

It took another decade for investigators to explore the use of guinea pigs as in vivo models of VZV infection. This exploration included ocular infections, such as keratitis, and latency in the trigeminal ganglion [15,16]. In analyses to investigate the ability of immunization to protect the eye from VZV infection, guinea pigs were immunized with VZV glycoproteins E and I (gE and gI), followed by inoculation of VZV into the vitreous of the eye, leading to chronic uveitis as well as involvement of the lungs and central nervous system (CNS) in the non-immunized animals. Later investigators failed to duplicate such extensive VZV infection of guinea pigs, although different routes of infection were utilized [17]; nevertheless, prior immunization of guinea pigs appeared to partially protect them from vitreal infection with VZV [18].

Several investigators used what were then newly developed molecular diagnostic techniques (e.g., polymerase chain reaction, PCR) to test the usefulness of guinea pigs as models for the study of VZV pathogenesis, clinical disease and antiviral therapy [2,3,4,19]. Using weanling animals, which were presumably more susceptible to infection, and hairless guinea pigs to enable ready detection of a rash, the investigators found that infections of the animals were more apparent if the VZV inoculum had first been adapted to growth in guinea pig cells. In all cases, the inoculum consisted of cells in which the virus was cultured; in no case was cell-free VZV or VZV-infected lymphocytes, which carry VZV during viremia, used to infect animals. The investigators were able to document cell-associated viremia and nasopharyngeal infection in weanling guinea pigs, although the animals did not become ill. Infection was successful when the inoculum was administered intranasally, subcutaneously and intracranially, whereas transmission of VZV to uninfected weanling animals in the same cage was limited when VZV was administered intranasally. Transmission of infection, however, was documented, not by observation of a transmitted illness, but by the seroconversion of rare exposed animals. A mild papular rash was noted in 5–70% of VZV-inoculated animals, but no vesicles characteristic of human varicella were observed. Hypothermia occurred in some animals and weight gain was depressed, but the animals were not otherwise ill. Overall, the guinea pigs were described as minimally symptomatic from VZV infection. Infections were most prominent in hairless animals, less in strain 2 guinea pigs and even less in outbred animals [2,3,4,9]. Pretreatment with cyclophosphamide, methotrexate and steroids to induce immunosuppression did not appear to enhance the severity of infection [17]. No VZV was isolated from the infected guinea pigs, although PCR amplification detected viral DNA in the skin and blood of animals inoculated with VZV. Latent infection with VZV was suspected in some of the VZV-infected guinea pigs, but this suspicion could not be verified because reactivation of VZV from latency could not be demonstrated. The main obstacle to success with the guinea pig studies in providing a clinical model for the study of VZV was thought to rest in a low multiplicity of VZV infection in these animals [17]. Subsequent guinea pig studies by Arvin and colleagues [20,21,22,23,24] were aimed at identifying the immunological events that follow infection and the varicella protein(s) that might serve in potential VZV vaccines; however, these investigators became interested in VZV pathogenesis, about which about they have learned a great deal from studies of the infection of human tissue implanted into mice with severe combined immunodeficiency (SCID-hu mice) [25].

### 2.3. Recent Studies of VZV in Guinea Pigs

Gershon and colleagues [26,27] originally used guinea pigs as a source of neurons to investigate lytic, latent and reactivating VZV infection in vitro. At that time, VZV was thought to exhibit a tropism for sensory neurons because the virus was known to establish latency in neurons of dorsal root (DRG) and cranial nerve (CNG) ganglia. Human DRG and CNG neurons can be obtained from cadavers or aborted fetuses, but neither source is ideal for laboratory experiments. Intrinsic primary afferent neurons (IPANs), thought to be analogous to the sensory neurons of DRG and CNG, are abundant in the enteric nervous system (ENS) [28,29], and guinea pig enteric neurons are readily isolated and cultured [30]. Thus, Anne and Michael Gershon tested the hypothesis that VZV would preferentially infect IPANs in neurons isolated from the guinea pig ENS. VZV, however, infected all enteric neurons in vitro, with no preference for IPANs. In humans, moreover, after an episode of varicella or inoculation with the live attenuated VZV vaccine, VZV establishes latency, not just in DRG and CNG, but also in ganglia of the autonomic nervous system, including those of the ENS and sympathetic nervous system [31,32,33,34,35]. The Gershons also noticed an interesting dichotomy in their cultures of guinea pig enteric neurons. When fibroblasts were present in the VZV inoculum, as they were when cell-associated VZV was used to infect cultures or when fibroblasts were included with enteric neurons at the time of isolation, the infection of neurons induced expression of VZV glycoproteins, caused rapid cell death, and thus appeared to be lytic [30,36,37], whereas use of cell-free VZV led to apparent virus latency. Neurons with apparent latency did not express VZV glycoprotein and survived for as long as cultures could be maintained. Interestingly, VZV could be reactivated from latency by expressing either VZV open reading frame (ORF) 61 or its orthologue in herpes simplex virus type 1 (HSV-1), infected cell polypeptide 0 (ICP0), in the infected neurons. Reactivation was evidenced by expression of VZV glycoproteins, death of the infected neurons within 48–72 hours, electron microscopic detection of gE-immunoreactive virions and infection of co-cultured human melanoma (MeWo) cells. This dichotomy suggested that the absence of ORF61 at the time of infection resulted in latent infection of neurons. While such a role for ORF61 awaits confirmation, the work with guinea pig neurons in vitro led to an investigation of whether VZV infects neurons of the human ENS in situ. Studies of the human gut suggested that VZV may be latent in the ENS of many individuals who have either experienced varicella or been vaccinated against it [33,34,35,36,37].

After a successful infection of isolated guinea pig neurons with VZV in vitro and the demonstration that VZV is a natural ENS pathogen, experiments were carried out to reevaluate in vivo VZV infection and latency in guinea pigs. A new two-step approach was used, involving confirmation that VZV established latency in the DRG, CNG and the ENS of guinea pigs after viremia induced by infusion of VZV-infected lymphocytes, followed by the administration of tacrolimus to reactivate the virus, a step based on the successful use of tacrolimus-induced immunosuppression to reactivate simian varicella virus in monkeys [38]. VZV was shown to infect guinea pig lymphocytes as well as human lymphocytes [39]. The infection of peripheral blood mononuclear cells was primarily of CD3-immunoreactive T cells. When given intravenously to guinea pigs, VZV-infected lymphocytes led to latent infection in virtually every neuron of the ENS as well as the DRG [39]. No visible illness resulted. VZV-infected animals survived at least six months, maintaining normal activity and gaining weight. Later, however, tacrolimus in combination with corticotrophin-releasing hormone (a molecular mimic of the effects of stress) caused VZV reactivation and a condition resembling a disseminated zoster [40]. Use of a GFP-expressing virus driven by the VZV ORF66 promoter (provided by Dr. Paul Kinchington) revealed patches of green fluorescence in the skin of the hairless guinea pigs upon illumination with a Wood’s lamp (Figure 1). The animals with reactivating VZV lost weight rapidly and were euthanized when weight loss exceeded 20–25% of the initial body weight. VZV DNA and transcripts were detected in the blood, colon, ileum, DRG, liver, pancreas, heart, skin and kidney. Interestingly, studies have shown that an injection of VZV into the skin of an immunocompetent guinea pig or the direct injection of cell-free VZV into the guinea pig’s ENS can also induce latent infection of enteric neurons. Small numbers of DRG neurons were found with retrograde tracers to project to both the skin and the gut [40]. Interestingly, ORF63 protein is detectable immunocytochemically in enteric neurons with latent VZV infection [40]. Although transcripts encoding ORF63 have been observed in latently infected human neurons, the expression of ORF63 protein is controversial. It is thus conceivable that VZV latency in guinea pig enteric neurons is not identical to that in human DRG or CNG with respect to expression of ORF63 protein. Recently, Depledge and colleagues [41] identified a spliced VZV mRNA antisense to VZV open reading frame 61 (VLT), a spliced transcript encoding a protein with late expression kinetics in lytically infected cells in vitro and in zoster lesions, which is expressed during latency in human neurons. VLT suppresses ORF 61 gene expression in co-transfected cells, suggesting a mechanism by which it may contribute to maintenance of viral latency. Analysis of ganglia from guinea pigs latently infected with VZV for VLT will provide additional insight into the guinea pig model.

### 2.4. Concluding Remarks on the Guinea Pig Model

Viremia inevitably occurs during varicella [42]. T-cells carry VZV through the circulation, enabling VZV to reach not only DRG and CNG, but also the ENS [33,34] and other autonomic neurons [31]. If VZV reactivates in neurons that project to the skin, the diagnosis of zoster is relatively straightforward. When VZV reactivates in neurons that lack cutaneous projections, the diagnosis of zoster is rarely even suspected. Visceral pain and obvious intestinal involvement may occur during varicella and, when it does, the manifestation is thought to be ominous [43,44,45,46,47,48,49]. The ability of VZV-infected CD3-immunoreactive guinea pig lymphocytes to transmit infection to guinea pig enteric neurons, which remains latent, may thus truly mimic human VZV disease. It is not clear why the infection transmitted by lymphocytes to enteric and DRG neurons is preferentially latent; the phenomenon may have evolved to foster the survival of the host and thus persistence of the virus. Enteric neurons may not be the primary target of VZV traveling from their site of proliferation in the tonsils to the skin, from which they can be disseminated to infect new hosts. However, destruction of the ENS would not serve VZV. An intact ENS is essential for life, and latency allows both the virus and the host to continue living. Once the ENS has been infected, however, VZV can and does reactivate there [35]. Minor localized enteric reactivations might even be useful in maintaining long-term immunity to VZV. The gut contains so many immuno-effector cells that low-level viral reactivation would likely be controlled before they produce extensive damage. However, if reactivation within the ENS is not controlled, enteric zoster, which can lead to lethal pseudo-obstruction or perforation of the bowel, can occur [35,40]. The guinea pig VZV model in which both latency and reactivation can be studied may provide novel insights into the still unknown mechanisms by which infected lymphocytes transmit viral latency to ENS neurons, the provocateurs of VZV reactivation within enteric neurons, the signs of enteric zoster and the potential contributions of enteric zoster to the burden of gastrointestinal illness. Finally, subject to availability of guinea pig-specific reagents, innate and VZV-specific adaptive immunity can be studied during VZV infection and reactivation.

## 3. Cotton Rat Model

The cotton rat has been used in models of numerous virus infections, including respiratory syncytial virus, influenza virus, parainfluenza virus, measles virus, hantavirus, adenovirus and herpes simplex virus. Infection of cotton rat fibroblasts with VZV results in productive infection (unpublished observations). Inoculation of cotton rats intramuscularly along the side of the thoracic and lumbar spine with cell-associated virus results in a latent virus infection, with VZV DNA in the DRG. A viral transcript that is expressed during latency in human ganglia, *ORF63*, is also expressed in cotton rat ganglia, while another transcript which has rarely been detected during latency in humans, *ORF40,* is usually not expressed in ganglia of latently infected cotton rats [50]. It is important to note that neither acute infection (varicella) nor virus reactivation (zoster) has been shown to occur in cotton rats after inoculation with VZV. Thus, while VZV DNA and RNA are detected in the ganglia of cotton rats and resemble the same viral nucleic acids seen in latently infected human ganglia, the possibility of an abortive infection in the ganglia of cotton rats cannot be excluded. Detection of VLT [41] in ganglia from cotton rats at later times following inoculation may help determine the status of infection.

The role of many VZV genes in latent infection has been studied in the cotton rat. Several virus genes, *ORF1*, *ORF2*, *ORF10*, *ORF13*, *ORF17*, *ORF21*, *ORF 32*, *ORF47*, *ORF 57*, *ORF61* and *ORF66,* have been shown to be dispensable for latent infection in cotton rats based on the infection of animals with VZV with deletions or stop codons in the genes [50,51,52,53,54]. While VZV mutants in most of these genes grow to wild-type levels in vitro, three exceptions are noteworthy. First *ORF21*, which is essential for growth in cell culture, was dispensable for the establishment of latency [54]; however, since cotton rats were inoculated with complementing cells expressing ORF21, the abundance of the ORF21 protein may have helped the virus to establish latency. Second, while VZV deleted in *ORF17* is severely impaired for replication in cell culture at 37 °C, the mutant virus established latency at levels similar to those of wild-type virus in cotton rats, which have a core temperature of 39 °C [51]. Third, while growth of VZV deleted for *ORF61* is severely impaired in the cell culture, the virus establishes latency at levels similar to those of the wild-type virus [52].

VZV with deletions or stop codons in other genes, *ORF4*, *ORF29* and *ORF63*, shows impaired latency and lower levels of viral DNA in DRG of cotton rats compared with animals infected with wild-type virus [55,56,57,58]. *ORF4* and *ORF29* are essential for growth in cell culture and can only be grown in cells expressing the proteins. In each of these experiments, VZV was passaged once in non-complementing cells, and animals were inoculated with non-complementing cells containing the virus. *ORF63* is the most abundant transcript in latently infected human ganglia [59], and its deletion results in impaired virus replication in culture. Cotton rats inoculated with VZV deleted for *ORF63* had levels of VZV DNA in their ganglia similar to those in animals inoculated with VZV expressing *ORF63* at three days after infection; however, at six weeks, both the percentage of latently infected animals and the viral load in the ganglia were reduced in the animals infected with VZV deleted for *ORF63* [55]. Use of VZV with different mutations in *ORF63* revealed a strong correlation between mutants impaired for replication in cell culture and those impaired for the establishment of latency [56].

The cotton rat model was used in developing a VZV mutant that was impaired for latency, but not for replication, in vitro [60]. VZV with deletions in portions of *ORF62* and *ORF63* and rearrangements in the 3’ portion of the genome was found to replicate to wild-type titers in the cell culture, induce high titers of neutralizing antibody in guinea pigs and was significantly impaired for latency in cotton rats. Thus, the establishment of latency with the mutant virus did not strictly correlate with virus replication in vitro. This VZV mutant could represent a safer vaccine, since it might be less likely to reactivate from latency.

Overall, the cotton rat model has several features that make it an attractive model to study VZV. Animals can be reliably infected with the virus, and a transcript expressed during latency in human ganglia (*ORF63*) is also expressed in cotton rat ganglia, while a transcript rarely expressed during latency in humans (*ORF40*) is also usually not expressed in latently infected cotton rats. Ganglia can be removed immediately after death to reduce the risk of postmortem virus reactivation, and the level of latent VZV can easily be quantified in ganglia. However, like all small animal models of VZV, infection of cotton rats does not result in acute disease (varicella) or virus reactivation (zoster), and unlike human ganglia, explanted ganglia from cotton rats do not undergo viral reactivation. In addition, compared with other small animal models, cotton rats are difficult to work with since they are very aggressive animals, immunologic reagents are much more limited for these animals than for mice, and fewer suppliers sell these animals; accordingly, this model has not been used for different studies, including innate adaptive immunity of VZV, by researchers other than those who originally reported its use.

## 4. SCID-hu Mouse Model

While the human-restricted nature of VZV has hindered the study of VZV infection in small animal models, the use of intact human (hu) tissue xenografts in mice with severe combined immunodeficiency, i.e., SCID-hu mice, has facilitated in vivo investigations of the molecular mechanisms of VZV pathogenesis [61,62]. VZV replication in human DRG xenografts recapitulates many aspects of ganglionic infection during initial infection and reactivation from latency. Replication is limited to the xenograft because of the host restriction, enabling long-term studies in the absence of VZV-specific adaptive immunity, which is lacking in SCID mice. Here, we highlight valuable insights into the molecular basis of neurotropism made using the SCID-hu DRG model to reveal VZV interactions with differentiated human neural cells examined in their DRG tissue microenvironment and to define viral determinants of neurotropism in vivo with recombinant viruses containing targeted mutations in VZV genes [25,63].

Briefly, DRG were dissected from fetal spinal tissue (18 gestational weeks, Figure 2, Panel IA), procured by Advanced Bioscience Resources (Alameda, CA) in accordance with state and federal regulations [61]. A single ganglion (~2 mm^3^) was surgically implanted under the left renal capsule of a sedated six-week-old male *scid/scid* homozygous C.B.17 mouse (Figure 2, Panel IB). Histologic evaluation of DRG xenografts after four weeks demonstrated the expected organotypic architecture, in which neuronal cell bodies and their encapsulating satellite glial cells (SGCs) resided within a network of axons and other supportive cells [61]. Endothelial cells within the DRG microvasculature expressed the human-specific cell marker PECAM-1 [61]. Human DRG neurons expressed neural cell adhesion molecule (NCAM) and synaptophysin. NCAM+ nerve bundles are observed extending throughout the DRG xenograft as well as into the murine renal tubule network. At 20 weeks, nociceptive and mechanoreceptive neurons, the major neuronal subtypes present in postnatal and adult human ganglia, were readily distinguished using peripherin immunoreactivity and RT97 immunoreactivity, respectively [64]. DRG xenografts were maintained in SCID mice for an extended period of time (>1 year). The capacity for human neurons to survive long-term and extend axons throughout the xenograft and into the mouse renal tissue indicate that growth properties of differentiated DRG neurons are intact despite the absence of orthotopic synaptic partners. VZV inoculation of DRG xenografts resulted in productive infection and was accomplished by direct injection of VZV-infected fibroblast cells into the exposed xenograft [61]. Importantly, intravenous administration of VZV-infected T-cells also resulted in DRG infection, supporting the concept that VZV gains access to sensory ganglia during the cell-associated viremia characteristic of varicella [60]. VZV-infected DRG were recovered 7–28 days after inoculation to assess acute replication and at >56 days to assess the latent phase, as described [61,62,64,65,66,67,68].

The acute replicative phase of DRG infection (7–28 days after inoculation) is characterized by extensive cytopathic changes, the fusion of neurons with their surrounding satellite cells to form polykaryons and the production of infectious virus in both neurons and satellite glial cells (SGCs) [61,66]. The histologic appearance of VZV DRG during the acute phase (Figure 2, Panel IC) is similar to descriptions of cadaver ganglia recovered from patients with zoster at the time of death [68]. As observed in ganglion neurons of zoster patients, VZV infection elicits dramatic changes in the DRG xenograft cytokine milieu, including increased production of cytokines associated with neuroinflammatory responses (MCP-1/CCL2, IL-1α, RANTES and IP10/CXCL10) [64]. Profiling VZV susceptibility of neuronal subtypes revealed that RT97+ mechanoreceptive neurons are more resistant to VZV replication compared with peripherin+ nociceptive neurons (Figure 2, Panel II) [64]. This restriction occurs after entry and ORF61 expression, but prior to IE63 protein expression, indicating that VZV replication is repressed very shortly after mechanoreceptive neurons are infected.

VZV and HSV-1 infection of DRG xenografts differ significantly, in that HSV-1 infects neurons only; the capacity of VZV to infect SGCs and fuse neurons and SGCs is unique [69]. Of note, neuronal subtypes that restrict HSV-1 and HSV-2 infection have been defined in a mouse ganglia model [70], suggesting that human alphaherpesviruses may have evolved tropism for skin-resident nociceptive nerve endings as a portal to neuronal ganglia. The state of VZV in DRG xenografts after the replicative phase resembles VZV latency as described in human cadaver ganglia obtained from seropositive patients without evidence of zoster at the time of death, i.e., viral genomes persist in the absence of infectious virus production. Viral transcription during this phase is highly restricted; ORF63 transcripts are present at low levels. This transition is unique to neurons within the DRG microenvironment, contrasting dramatically with the progressive lytic infection observed in skin and T-cell xenografts, and occurs despite the absence of adaptive immune responses [25]. The extent to which VZV genes are transcribed and translated during latency has been challenging to resolve using cadaver ganglia because activation of the entire VZV transcriptome, in particular ORF63 transcript levels, was shown to increase in proportion to the postmortem interval [71]. In addition, endogenous antibodies directed against blood group A1-associated antigens present in animal sera and ascites-derived antibody preparations used to detect IE63 protein produce a staining artifact in cadaver ganglia that can be misinterpreted as VZV-specific staining due to neuronal A1 expression [72]. Experiments using DRG xenografts to investigate VZV neurotropism as well as to examine cadaver ganglia after adsorption of sera or ascites against blood group A1 erythrocytes have demonstrated that VZV protein expression is not a characteristic of latency (Figure 2, Panel I) [72,73]. Investigation of VZV reactivation in the SCID-hu DRG model has been challenging due to the absence of a marker to identify latently infected neurons. The presence of VLT [41] has not been assayed in the SCID-hu model. DRG xenograft experiments also revealed that subnuclear domains known as promyelocytic leukemia protein nuclear bodies (PML-NBs), or ND10 bodies, serve as an intrinsic defense (Figure 2, Panel III). Large PML-NBs, consisting of PML fibers that sequestered newly assembled nucleocapsids (NC) in spherical cages, were identified in neurons and satellite cells [74]. Preventing nuclear egress of NCs was identified as a basic cytoprotective function of PML in DRG as well as skin.

Analyses of VZV determinants of neurovirulence have focused on strain comparison (vaccine and clinical isolates) and the use of VZV recombinant viruses generated by overlapping cosmid or bacterial artificial chromosome (BAC) systems. Importantly, infection of DRG xenografts using live attenuated varicella vaccine virus demonstrated that vaccine Oka (vOka) exhibits the same pattern of acute replication followed by viral DNA persistence as two clinical isolates, parent Oka virus and VZV-S [61]. The experiments suggest that vOka attenuation by serial passage in cultured cells does not alter infectivity for neurons or the capacity to establish a transcriptionally repressed latent state, consistent with the clinical evidence that vOka can establish latency and reactivate to cause zoster. Experiments to identify VZV determinants of neurovirulence using VZV recombinant viruses have focused on VZV ORF68-encoded glycoprotein gE and ORF67-encoded gI, which are type I membrane proteins that participate in cell–cell spread and secondary envelopment [75]. Those experiments demonstrated that gE interaction with the insulin-degrading enzyme, identified as a possible cellular entry receptor, is not required for neurotropism, whereas the gE interaction with gI to form heterodimers as well as specific gI functions needed for virion assembly and egress are critical for VZV replication in DRG in vivo [65]. Notably, disrupting gE–gI binding resulted in chronic low-level replication and extensive destruction of DRG cells, indicating that preservation of the gE–gI interaction is necessary for the transition to VZV persistence in neurons.

In summary, the SCID-hu DRG model has enabled substantial progress in the understanding VZV virulence factors and neurotropism. That VZV exhibits tropism for both neurons and SGCs and can cause fusion of these unique cell types distinguishes VZV from HSV neuropathogenesis, and helps to explain the often prolonged recovery from zoster and the occurrence of post-herpetic neuralgia. While this review is restricted to neurotropism, it should be noted that SCID-hu model is useful for studies of innate immune response. Although many questions remain, further advances are expected to provide better insight into the clinical manifestations of varicella and herpes zoster.

### Application of the SCID-hu Mouse Model to Study VZV ORF7 Neurotropism

VZV encodes 70 unique ORFs. In a genome-wide mutagenesis study, each of the VZV ORFs was individually deleted using VZV BAC technology [76]. That study showed that 52 of 70 ORFs are essential or critical and 18 of them are dispensable for viral replication in MeWo (human melanoma) cells [77]. To identify the factors required for VZV neuronal invasion, latency and pathogenesis, the 18 non-essential ORF mutants were first used to infect human differentiated neuroblastoma and embryonic stem cell-derived neurons. Of the mutants screened, only the ORF7 mutant (7D) failed to grow and spread in the neurons [78], suggesting that ORF7 encodes a neural tropic factor.

To further test ORF7 neurotropism in more physiologically relevant systems, the SCID-hu DRG xenograft mouse model was used [61]. A human fetal DRG xenograft (Figure 2A,B) was implanted under the kidney capsule of a SCID mouse. Four weeks post-xenotransplantation, the DRG was surgically exposed and directly injected with 100 PFU of the wild-type (WT), ORF7 deletion mutant (7D) and 7D rescue (7R) VZV (Figure 3D). Since the viruses contained the luciferase marker, the viral replication in vivo was measured for 10 days using the In Vivo Imaging System; no luminescent signal was detected in the 7D-infected SCID-hu DRG xenograft (Figure 3C,D). Importantly, 7R grew at the same rate as did the WT, demonstrating that the defective phenotype of 7D is fully restored by rescuing ORF7. Histochemical analysis and immunostaining of the harvested and fixed infected DRG grafts (Figure 3E) revealed abundant VZV gE antigen in WT- but not in 7D-infected neurons or surrounding satellite cells. For WT, the pervasive presence of gE antigen indicates active viral replication. Vacuolar structures might indicate the formation of neuron-satellite cell complexes or satellite cell polykaryon (arrows in Figure 3E). In contrast, 7D was unable to propagate within the implant, as suggested by the absence of gE antigen, such that the appearance of the DRG is that of a normal tissue. As shown in Figure 3F, fluorescence in situ hybridization of DRG xenografts using a whole-VZV genome fluorescently labeled probe revealed the widespread presence of viral DNA in the nuclei of WT-infected neurons (red arrowheads) and satellite cells (white arrowheads). Clustering of viral DNA in specific intranuclear replication centers is an indication of productive replication (bright green fluorescence). In contrast, no viral DNA was detected in the 7D-infected sample (Figure 3E), consistent with the results of the immunohistochemistry analysis. These results indicate that the absence of ORF7 from the viral genome prevents the VZV spread in human DRG tissue.

## 5. Simian Varicella Virus (SVV) Infection in Non-Human Primates (NHP)

### 5.1. Molecular Aspects of SVV

Simian herpesviruses were identified as the causative agent responsible for epizootic outbreaks of monkey chickenpox in Old World monkeys [79]. Virus isolates were initially named for the geographical location of the epizootic and/or the monkey species (i.e., Liverpool vervet virus, Medical Lake macaque virus and Delta herpesvirus). Subsequent molecular analyses of viral DNA derived from these distinct outbreaks confirmed that the epizootics were caused by the same infectious agent, which was named simian varicella virus (SVV) [80], which is now classified as *Cercopithecine herpesvirus 9* within the *Varicellovirus* genus of alphaherpesviruses.

In the laboratory, SVV is grown in cell cultures of simian origin, usually African green (Vero, BSC-1, CV-1) or rhesus (LM-MK2) monkey kidney cells [81]. Infected cell monolayers exhibit a cytopathic effect, with multi-nucleated cells. Electron microscopy reveals viral nucleocapsids within the cell nucleus [79]. In addition, abundant degraded viral particles are detected within cytoplasmic vacuoles, apparently caused by virion sensitivity to cellular lysosomal enzymes. Consequently, SVV-infected monolayers generate a low titer of infectious virus (10^2^–10^4^ plaque-forming units per ml) compared to many other herpesviruses. Such is also the case for VZV grown in vitro [82]. SVV and VZV mature as cell-associated viruses, since relatively little infectious viruses are released from infected cells into the culture medium.

SVV virions have typical herpesvirus morphology containing at least 30 protein species ranging in size from 16 to >200 kD, including at least six envelope glycoproteins (46–115 kD) [83]. SVV and VZV proteins share extensive antigenic cross-reactivity, as demonstrated by the cross-reactivity of sera derived from SVV-infected animals and from herpes zoster patients with VZV and SVV antigens, respectively [84], and by the finding that monkeys experimentally immunized with VZV are protected from acute simian varicella infection following an SVV challenge [85].

Although SVV and VZV are two different viruses, their genomes are similar in size, structure, genetic content [86,87,88] and pattern of disease produced in their respective hosts. SVV DNA consists of 124,785 base-pairs (bp), 99 bp shorter than VZV DNA, with a 40.4% G + C content comparable to that of VZV DNA (46.0%) [89]. Similar to VZV DNA, the SVV genome is composed of a 104-kbp unique long (UL) component bracketed by a 64-bp inverted repeat sequences and a short (S) component composed of a 4.9-kbp unique short (US) region bracketed by 7.6-kbp inverted repeat sequences. SVV DNA encodes 69 distinct SVV ORFs, each sharing 27–75% amino acid identity to corresponding VZV genes [90]. Like VZV DNA, the SVV genome includes three ORFs (62–64) that are duplicated within the S segment. However, a VZV ORF 2 homologue is missing in the SVV genome. SVV and VZV genomes differ only in the 3-kbp sequence at the left end of the genomes (Figure 4) [91]. The function of SVV ORF B (VZV ORF 0 or S/L) is unknown. Based on their homology to VZV ORFs, SVV ORFs 1 and 3 probably code for membrane and virion assembly proteins, respectively. SVV does not contain ORF 2. SVV ORF A, not present in VZV is homologous to the C-terminal end of SVV ORF 4, which codes for an immediate early gene and is a transactivator.

Like other herpesviruses, SVV gene expression during viral replication is understood to be coordinately regulated into immediate early (IE), early and late phases. SVV *ORF62* encodes a major transactivator protein (IE62), which is expressed early in infection and induces transcription of SVV IE and early genes [92]. SVV IE *ORF61* and *63* also encode transcriptional activators [93]. SVV early genes, which are involved in viral DNA replication, include deoxyUTPase (*ORF8*), ribonucleotide reductase (*ORF19*), thymidine kinase (*ORF36*) and uracil DNA glycosylase (*ORF59*) [94,95]. The viral capsid proteins and several envelope glycoproteins, including SVV gB (ORF31), gC (ORF14), gE (ORF68), gH (ORF37) and gL (ORF60), are expressed during the late phase of viral replication [89,96,97,98].

During acute varicella infection of NHP, most of the SVV genome is transcribedd, as confirmed by the transcriptome analysis of bronchial alveolar lavage (BAL) cells and peripheral blood mononuclear cells (PBMCs) [99]. SVV IE *ORF63* and *ORFs41* and *49*, which encode structural proteins, are abundantly expressed during an acute disease. In contrast, SVV gene expression is restricted during viral latency. An SVV latency-associated transcript (LAT), identical to VLT [41], is predominately expressed in cells of neural ganglia derived from latently infected nonhuman primates [99,100]. Similar to VLT, SVV LAT is transcribed anti-sense to the SVV *ORF61* transactivator gene. Although SVV LAT was discovered first, it is not as well characterized as VLT. The simian varicella model provides a useful approach to further explore the role of VLT in viral latency.

SVV infection of non-human primates (NHP) provides the only experimental animal model that closely simulates human varicella and studies involving innate and adaptive immunity. The study of SVV molecular biology and pathogenesis is facilitated by the ability to generate gene-specific viral mutants. Initially, an SVV cosmid recombination system for generating SVV mutants was used [101], in which SVV DNA fragments (30–45 kb) collectively composing the entire SVV genome were cloned into four overlapping SVV cosmids. Transfection of these cosmids into Vero cells and homologous recombination generated infectious SVV. Site-specific mutations introduced into individual cosmids have facilitated the generation of SVV gene mutants, including SVV gC^–^, ORF61^–^, UDG^–^ and dUTPase [93,101,102]. Similarly, the cosmid system has been used to construct recombinant SVVs expressing foreign genes such as those encoding simian immunodeficiency virus (SIV) gag and env antigens [103]. More recently, the entire SVV genome has been cloned into a BAC, permitting propagation of the SVV DNA in *Escherichia coli* and facilitating the generation of SVV mutants and recombinant viruses [104,105,106]. SVV derived from this SVV-BAC is pathogenic upon infection of infected rhesus monkeys and establishes latent infection in neural ganglia [107].

### 5.2. Primary SVV Infection and Latency

Despite being a different herpesvirus infecting a different host, SVV is pathogenic in NHP causing an acute respiratory infection with varicella (chickenpox) and later establishment of latency in sensory and enteric neurons, similar to disease produced by VZV infection in humans. Multiple species of NHP are susceptible to natural SVV infection [79,108,109]. Further development of NHP as animal models using experimental intratracheal, intrabronchial or natural aerosol routes for SVV inoculations demonstrated distinct species-specific patterns of disease [110,111,112,113,114]. Experimental infection of African green monkeys (AGM) leads to severe, sometimes fatal hemorrhagic disease (Figure 5A), whereas Indian rhesus macaques (RM) and cynomolgus macaques develop a less severe disease that is more similar to humans [110,111,112,113,115]. SVV inoculation in Chinese RM was shown to develop immune responses without a skin rash [116]. Subspecies differences in disease course between Chinese and Indian RM have also been observed in SIV infections [117]. Similarly, unlike macaques, non-macaques infected with herpes B virus can suffer fatal encephalitis [118].

Experimental respiratory inoculations of SVV in AGM results in an infection of epithelial, myeloid and lymphoid cells. A Virus then enters the bloodstream infecting PBMCs to cause a viremia, with further hematogenous dissemination to the skin, lymph nodes, spleen and other organs [114,119]. Peak SVV-infected cell numbers are seen in BAL at 3–9 days post-infection (dpi), while SVV viremia is observed as early as 3–4 dpi in AGM and RM [112,120,121], with infection of all major PBMCs including T- and B-lymphocytes, natural killer cells, monocytes and dendritic cells in AGM [114]. In RM, in some instances, very low viremia was seen at 7 dpi with robust SVV replication in lungs and the virus did not infect B cells [122]. Strong pulmonary innate immune cytokine and chemokine responses during acute infection are also seen in RM correlating with low viremia [123,124]. RNA sequencing studies on tissues from SVV-infected RM have demonstrated that virus spreads rapidly through the lungs, replicating both in the immune cells and parenchyma [125]. Acute infection is characterized by significant immune infiltration and reduced expression of genes important for lung function, indicative of lung injury and viral pneumonia. This is followed by increased expression of genes important for host defense, which correlates with viral clearance. Finally, the expression of growth factors and cell cycle-associated genes increases to usher in the repair phase [125].

The first signs of clinical disease, seen frequently in AGMs and sometimes in macaques, are a fever and enlarged lymph nodes at 6–14 dpi [114]; Traina-Dorge, unpublished observation]. At this time and coincident with peak viremia, animals develop a limited to widespread generalized macular, papular and vesicular rash, with an occasional severe hemorrhagic rash in AGMs [112,113,126,127]. Viral lesions in the rash are widespread over the body, including the torso, abdomen, appendages, face and neck. Mild hepatitis may also be present at 9–10 dpi in AGMs and RMs, with elevated liver transaminase levels in the blood [38,112].

At 7–14 dpi, a robust adaptive immune response develops, characterized by the presence of both binding and neutralizing antibodies and the detection of antigen-specific CD4 and CD8 central and effector memory T-cells in both BAL and PBMC samples [79,113,114,123,128,129]. Studies using the RM model clearly indicate a critical role for CD4 T-cell immunity in resolution of acute SVV infection [130] and the establishment of latency [131], as well as reactivation [132,133]. As early as 3 dpi, SVV probably traffics to the ganglia by hematogenous transport. Ganglionic infection is indicated by the detection of SVV DNA and antigens in neurons, although virus-induced cytopathology is absent [79,134,135,136]. Infiltration of T-cells [122] and SVV-infected T-cells containing SVV antigen [114] surrounding the sensory neurons have been observed, supporting their role in viral transport to ganglia. SVV infection of ganglia is accompanied by robust changes in gene expression [122]. Differentially expressed genes detected at 3 dpi include components of the troponin complex, which play a critical role in axonal transport, as well as several interferon stimulated genes indicative of an innate antiviral response. Upregulation of genes important for T- and NK cell recruitment is seen at 7 dpi, followed by significant downregulation of genes important for neuronal function and nervous system development, which lasts up to 100 dpi, long after cessation of viral replication. After acute infection, virus-infected T-cells expressing the gut-homing receptor α4β7 integrin were detected in mesenteric lymph nodes that drain into the gut, supporting the role of T-cells in viral transport to enteric neurons. SVV RNA has also been shown in enteric neurons of latently infected monkeys, similar to observations in humans with VZV infection [33,137].

In RM, a virus is cleared from the peripheral organs within 7 dpi, with establishment of latency [122,138]. In the AGM model, after intratracheal inoculation, SVV DNA and RNA persist in lung and liver for up to 10 months, in ganglia for up to 12 months and in circulating T-cells for years after infection [111,139]. During SVV latency, the viral gene expression program is highly restricted, with only 12 ORFs detected in the ganglia of latently infected RM [99]. Among these transcripts, an ORF61 antisense transcript was 5–9 times more abundant than a sense transcript [100,113]. As recently shown for VZV [41], this antisense transcript is thought to abrogate ORF61 expression in order to promote the maintenance of SVV latency. Interestingly, deletion of ORF61 from the SVV genome did not impair establishment of SVV latency, suggesting that the absence of both ORF61 and its antisense transcript masks their relative functions in viral latency [139].

Accumulating evidence points to the important role of T-cells in SVV dissemination, establishment and maintenance of latency. First, it was shown that T-cells in the BAL are susceptible to and can support SVV viral replication in vivo [140]. Moreover, memory CD4 and CD8 T-cells infiltrate ganglia as early as 3 dpi, which correlates with the detection of viral loads and before the development of cell-mediated immunity [122]. A recent study reported that SVV infection causes large changes in the expression of genes that play a role in cell cycle, cellular metabolism and immune processes in T-cells isolated ex vivo from BAL samples collected from RM at different dpi [140]. Together, these data indicate that T-cells are susceptible and permissive SVV targets, with a critical role in viral dissemination to the ganglia.

### 5.3. SVV Reactivation in NHP

A significant complication of aging is the reactivation of latent VZV, manifesting as zoster, post-herpetic neuralgia (PHN), pneumonitis, blindness, stroke, myocardial infarction and other multi-organ diseases. VZV latency is controlled largely by cell-mediated immunity and reactivation is due to its decline, disruption or loss [141]. In humans, immunosuppression due to radiation therapy or treatment with immunosuppressive drugs and/or steroids induces reactivation of VZV, causing severe and sometimes fatal zoster, neuritis, PHN and other complications [142,143,144]. VZV does not reactivate in any small animals used to study establishment of latent infection. However, SVV reactivates in immunosuppressed pigtail, cynomolgus and RM and AGMs [108,115,145,146,147].

The NHP model of varicella reactivation has been successfully developed in SVV-infected cynomolgus and RM and AGMs [120,127,142] using immunosuppression to disrupt cellular immunity through a combination of X-irradiation, tacrolimus and prednisone (Trio XTP) to induce reactivation of latent SVV in macaques (Figure 5B). In cynomolgus macaques, zoster developed after immunosuppression, with SVV-antigen detected in a skin rash, sensory ganglia and the lungs. SVV glycoproteins were also found in non-ganglionic tissues in the absence of a rash in some of the treated monkeys and in one untreated monkey subjected to the same stress of transportation [148], indicating subclinical reactivation.

In another study, more monkeys developed zoster when treated with tacrolimus alone rather than with both tacrolimus and irradiation [127]. As mentioned above, tacrolimus to induce varicella reactivation was also used in the guinea pig model [40]. Infiltration of T-cell clusters around ganglionic neurons was seen in cynomolgus macaques that developed zoster after tacrolimus treatment. CD8 T-cells were more prevalent than CD4 T-cells around neurons. In addition, CXCL10 RNA but not SVV ORF61 RNA co-localized with the T-cell clusters. CXCL10 RNA-positive cells were more abundant within two months after zoster compared to the later times [149]. The T-cell clustering may be mediated by CXCL10 and parallels the observation in autopsied ganglia of humans after zoster [150].

SVV reactivation was observed in infected RM treated with Trio-XTP [120]. Zoster rash was also observed in an untreated monkey subjected to the same stress as treated monkeys. SVV antigens were detected in skin, lung, lymph nodes and sensory ganglia. Viral antigens in the skin were localized in the epidermis, around hair follicles and sweat glands; in the lung, primarily in the alveolar wall; in the lymph nodes, colocalized mostly with macrophages, dendritic cells and, to a lesser extent, T-cells; and in the sensory ganglia, in neuronal and non-neuronal cells. Unlike virus antigens, only minimal levels of SVV DNA were detected in the tissues.

Similar to observations in cynomolgus macaques, numbers of circulating PBMC were significantly reduced in RM within a week after the start of the Trio-XTP treatment [128]. Flow cytometric analyses of blood samples after treatment revealed reduced numbers of T-cells, including naïve, memory and effector CD4 and CD8 T-cells. These values remained low for days to weeks after initiation of treatment. However, five weeks before the appearance of zoster, the percent of naïve CD8 T-cells spiked, while the abundance of memory and effector CD8 T-cells remained low in the treated, but not in the untreated, monkeys. CD8 T-cells in the untreated monkey were also reduced after transportation to the radiation facility, but not to the same extent as the treated animal. SVV reactivation may have been initiated by the lack of viral control by memory and effector cells. Two weeks before zoster, all T-cell subsets were increased, possibly as a response to virus reactivation as confirmed by the detection of virus antigen in tissues at necropsy [120]. Programmed death receptor (PD-1), a marker of T-cell activation, was also increased in all CD4 and CD8 T-cells during zoster compared to latency, suggesting an antigen-driven response. PD-1 is also a marker of T-cell exhaustion that has been found on T-cells of patients with chronic viral infections and may signal T-cell decline and exhaustion at the time of zoster [151]. SVV-specific responses could not be confirmed due to the lack of availability of virus-specific reagents for flow cytometry. Intracellular cytokine responses might have confirmed an association with virus reactivation.

Assessment of cytokine and chemokine levels in serum after immunosuppression and during SVV reactivation showed a strong increase in the pro-inflammatory mediators MCP-1, eotaxin, IL-6, IL-8, MIF, RANTES and HGF, but lower levels of the anti-inflammatory mediator IL-1Ra in the circulation [124]. The influx of these strong inflammatory cytokines and chemokines is consistent with an acute inflammatory response to the reactivated virus, as observed in zoster patients as early as nine days post-zoster [152].

Subclinical reactivation of latent SVV in RM was demonstrated using thymectomy along with either CD4 or CD8 depletion [133]. Following treatment, initially and for the first year thereafter, no SVV DNA was detected in the whole blood nor was any rash seen in the animals. However, within 1–2 months after a stressful event (moving the monkeys to a different room), SVV DNA was detected in whole blood in some of the CD4-depleted and CD8-depleted monkeys, in the absence of a zoster rash [133]. More recently, clinical reactivation of latent SVV was shown in RM after CD4-depletion, in which circulating CD4 T-cells were reduced by more than half within a week and remained low for two months; a zoster rash was seen in one animal with the most extensive CD4-depletion at seven days post-treatment and in the other treated RM at 10–40 days post-treatment, with recurrent zoster observed in one animal [132]. Although the untreated control retained normal CD4 levels, it also developed zoster. This animal had very low SVV-specific antibody levels suggesting overall low virus-specific immunity. SVV reactivation was confirmed by the presence of SVV antigens in multiple tissues and virus DNA in lung and lymph nodes. Although the reason for the contrasting results in the two studies [132,133] is unclear, both studies suggest an important role for CD4 T-cell immunity in controlling varicella virus latency. Together, despite being an expensive model, SVV infection in NHP continues to be an extremely valuable animal model, expanding our understanding of the mechanisms of neurotropism and pathogenesis of varicella virus. Recently, we have shown that SVV DNA in saliva can be used to monitor acute infection in RM [153]. Future studies using the NHP model will include PHN, shedding of virus in saliva during reactivation, giant cell arteritis and corneal reactivation disease.

## 6. Conclusions

After primary infection, VZV establishes a life-long latent infection in neurons of the cranial nerve, dorsal root and autonomic ganglia, and reactivates later to produce several neurological disorders. It has been a challenge to study VZV neurotropism using an animal model since the virus causes diseases only in humans. In this review, we have described most of the animal models that have been adopted to study VZV neurotropism. The advantages and the disadvantages of each of the models are listed in Table 1. Attempts to infect small animals with VZV include cotton rats, guinea pigs and SCID-hu mice. Some of the important criteria to be met by any animal model are: virus replication in host cells, induction of varicella; ganglionic infection followed by the establishment of latency and the ability to reactivate to produce zoster. During VZV latency, virus nucleic acids are seen only in ganglia, virus transcription is restricted and virus is present only in neurons and has the ability to reactivate. Recent developments using guinea pigs for experimental VZV infection have shown that in the absence of primary disease, virus enters the enteric ganglia and becomes latent, suggesting the usefulness of this model to study latency and reactivation. One limit of guinea pig studies, which might help to explain why they have not been more widely utilized, is that the infection of animals do not mimic primary varicella. Earlier studies have shown that experimental infection of rats and mice with VZV does not produce varicella, but at later times, the virus is found in ganglionic neurons as well as in non-neuronal cells and in extraneural tissues. The cotton rat model has been used not only to identify the virus genes required for ganglionic infection, but also to show restricted transcription of VZV genes in ganglia weeks after experimental inoculation. Foot-pad inoculation of VZV into Sprague-Dawley rats demonstrated incomplete infection but was sufficient to produce pain behaviors that involved changes induced in neuronal populations [154]. The SCID-hu model has been a good alternative in vivo model to study VZV neurotropism as well as the role of specific virus genes, such as ORF7 in ganglionic infection. SVV infection of NHP has provided extremely useful information regarding varicella pathogenesis, latency and reactivation, despite differences between VZV and SVV with respect to: Reactivation, when VZV dermatomal rash vs. SVV whole-body rash; the species dependency of experimental SVV infection and differences in the 3-kb sequence at the left end of the virus genomes. Overall, the future of research on VZV neurotropism and pathogenesis rests on the extension of the existing important animal models to determine critical virus-host immune and neuronal cell interactions in different tissues at multiple times post-reactivation, studies that are currently otherwise impossible in humans.

## Figures and Tables

**Figure 1 viruses-11-00502-f001:**
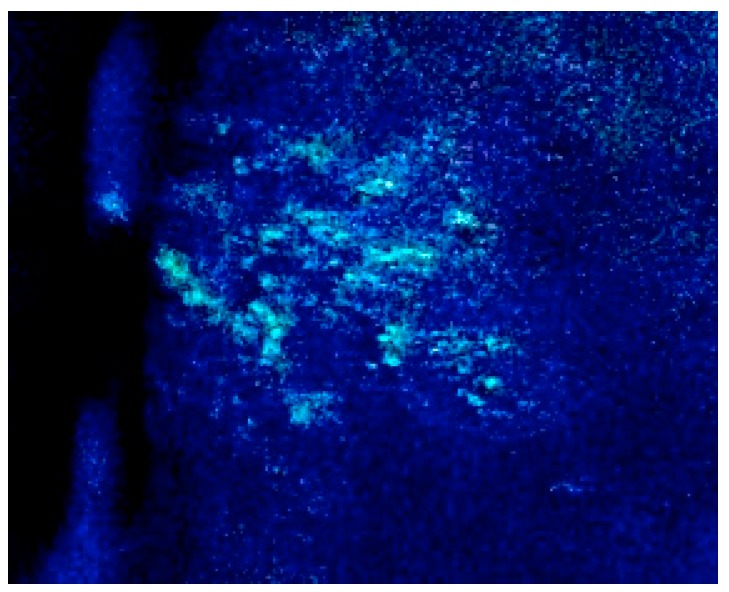
Wood’s lamp illumination of guinea pig skin after reactivation of the varicella-zoster virus (VZV). A hairless guinea pig was inoculated with lymphocytes infected with VZV modified to express GFP under the control of the promoter for open reading frame (ORF) 66. One month later, the animal was treated with tacrolimus to induce immunosuppression and with corticotrophin-releasing hormone to mimic stress. Patches of green fluorescence appeared on the skin, simultaneously with severe weight loss.

**Figure 2 viruses-11-00502-f002:**
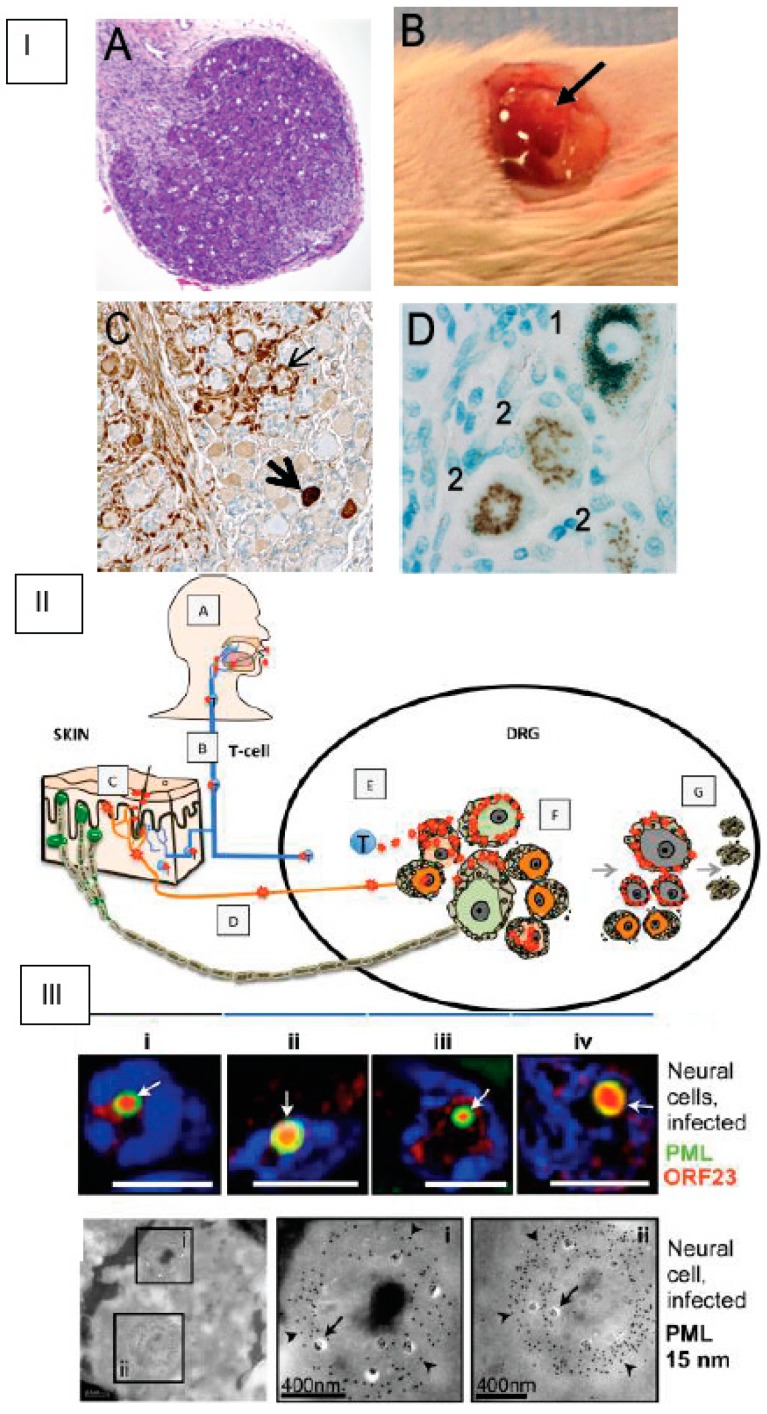
Varicella-zoster virus (VZV) infection of SCID-hu dorsal root ganglia (DRG) xenografts. Panel I. (**A**) Tissue section of fetal gestational DRG (18 gestational weeks) prior to implantation, stained with hematoxylin and eosin. (**B**) Appearance of DRG xenograft at four weeks post-transplant (arrow). (**C**) VZV-antigen immunohistochemistry in a tissue section of VZV-infected DRG during acute infection (brown signal indicates VZV protein in neurons (thick arrow) and satellite cells (thin arrow)). (**D**) Tissue section from a human cadaver DRG after VZV-antigen immunostaining, where numbers denote two different staining artifacts that occur when staining cadaver DRG tissue sections: 1, lipofuscin reactivity, which appears green upon counterstaining with azure blue; and 2, reactivity of blood group A1-associated antigens. Panel II. Primary VZV infection in respiratory epithelial cells (**A**) enables T-cell-mediated spread (**B**) to skin (**C**), followed by retrograde axonal transport to DRG neurons (**D**) or direct T-cell-mediated spread to DRG (**E**), which facilitates satellite glial cell (SGC) infection. VZV infects both nociceptive (small, orange) and mechanoreceptive (large, green) neuron cell bodies, but replication is severely restricted in mechanoreceptive neurons. If neuronal replication is uncontrolled, SGC infection facilitates spread to neighboring non-neuronal satellite cells (**F**) and associated neuronal cell bodies, amplifying VZV retrograde transport to skin. SGC infection also contributes to neuronal cell loss, resulting in satellite cell microproliferations (nodules of Nageotte; **G**). Panel III. Upper: Neural cells have large ring-like promyelocytic leukemia (PML) cages (green, white arrows), where ORF23 capsid protein localizes (red). Lower: PML-specific immunogold-labeling of ultrathin cryosections shows that PML cages (PML, 15 nm; arrowheads) sequester VZV nucleocapsids (arrows) in infected neural cells. Areas in black squares (**i** and **ii**) are shown at higher magnification in right panels.

**Figure 3 viruses-11-00502-f003:**
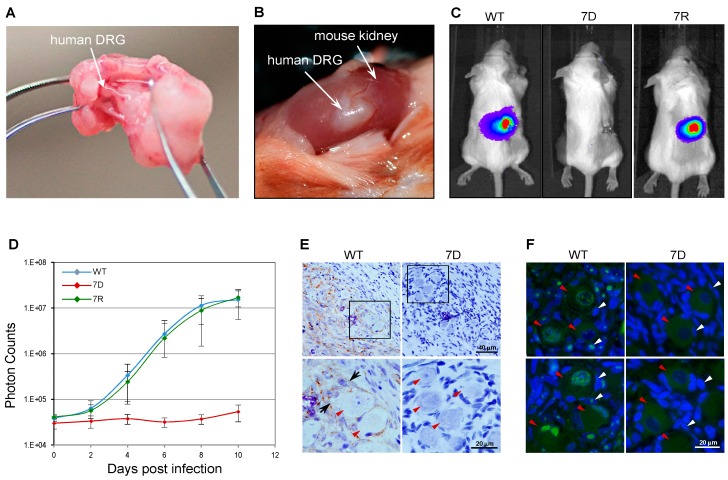
ORF7 is essential for viral replication in human neurons in vivo. (**A**) Human dorsal root ganglia (DRG) isolated from the fetal spinal cord. (**B**) Human DRG were implanted under the kidney capsule of SCID mice and four weeks later, infected with 100 PFU of cell-free wild-type (WT), the ORF7-deleted mutant (7D) or the 7D rescue (7R) VZV, respectively. (**C**) The bioluminescence signal recorded in SCID mice every other day after intraperitoneal administration of D-luciferin for 10 days. Images at 10 days post-infection (dpi) are shown. (**D**) The total photon counts from each sample (five animals/viral sample) were quantified, analyzed and used for generating growth curves. (**E**) At 10 dpi, WT- and 7D-infected DRG harvested from SCID mice were stained for gE (brown) and counterstained with hematoxylin (blue). Lower panels represent a higher magnification of selected areas of the upper panels. Red arrowheads indicate neurons. Vacuolization and possible neuron-satellite cell aggregates (black arrows) were observed in the WT-infected sample, correlating with productive replication and widespread presence of viral gE antigen (brown), while 7D samples retained normal DRG morphology and no gE was detected. (**F**) Fluorescence in situ histochemical staining of the infected DRG samples using a VZV genomic DNA probe (green) revealed intranuclear signals from nuclei of both neurons (red arrowheads) and adjacent satellite cells (white arrowheads) in WT-infected tissue, indicative of active viral DNA replication within intranuclear replication centers; no 7D DNA was detected in any of the samples tested at the experimental endpoint.

**Figure 4 viruses-11-00502-f004:**
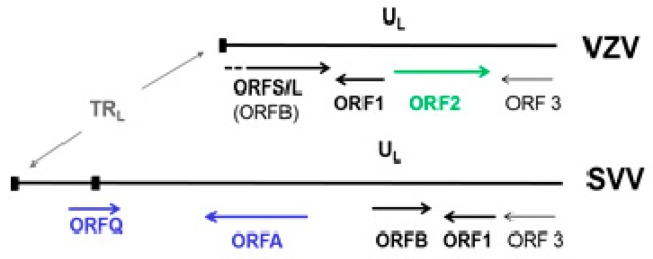
Sequence organization at the left ends of VZV and simian varicella virus (SVV) genomes. TR_L_—terminal repeat sequences; U_L_—leftmost part of the unique long segments. SVV ORF B is homologous to VZV ORF S/L. ORFs A and Q (blue) are present in SVV but not VZV, whereas ORF2 (green) is present in VZV but not SVV.

**Figure 5 viruses-11-00502-f005:**
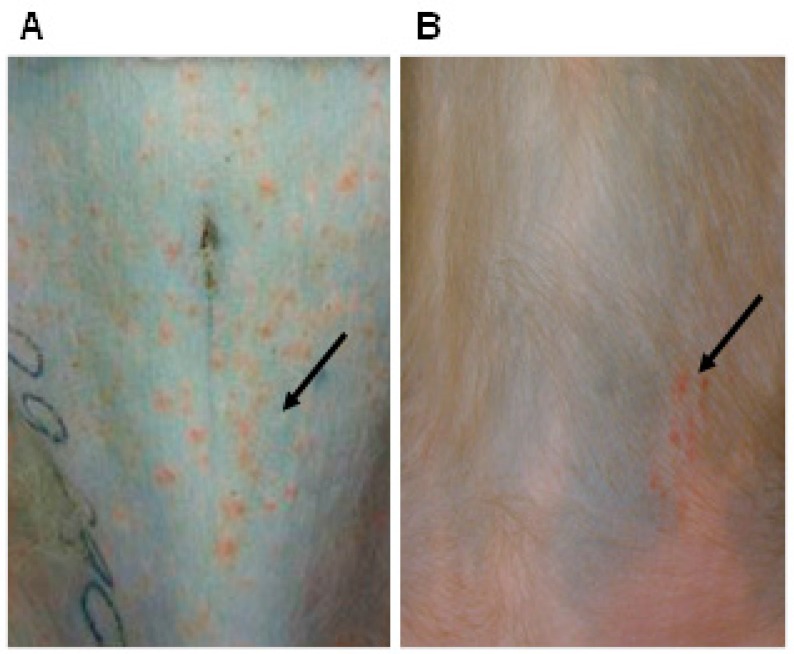
Varicella and zoster rash in non-human primates (NHP). (**A**) A hemorrhagic varicella rash in an African green monkeys (AGM), 13 days post-intratracheal inoculation and (**B**) a zoster rash in a latently infected rhesus macaques (RM), seven weeks after immunosuppression. Arrows indicate the location of rash on the skin.

**Table 1 viruses-11-00502-t001:** Evaluation of animal models of VZV neurotropism.

Animal Model	Advantages	Disadvantages
**Guinea pig**	VZV infects guinea pig cells and can proliferate. VZV infects guinea pig enteric neurons in vitro and can establish latency if the inoculum contains a low multiplicity-of-infection in the absence of non-neuronal cells. VZV can be reactivated from latency in guinea pig enteric neurons in vitro by expressing ORF61. Intravenous inoculation of VZV-infected guinea pig or human lymphocytes infects guinea pigs and establishes latency in most, if not all, enteric and dorsal root ganglion neurons; animals remain symptom-free for months. A combination of immunosuppression with tacrolimus and a simulation of stress with intravenous corticotrophin-releasing hormone can reactivate VZV from latency. Guinea pigs are less expensive than NHP. Ganglia can be removed immediately after euthanasia, to avoid concerns about postmortem reactivation.	Guinea pigs do not display typical varicella. Reactivation elicits a severe disseminated syndrome, like disseminated zoster in humans, necessitating that the animals be euthanized. Guinea pigs have a long gestation period and thus are difficult and expensive to breed, complicating studies of congenital VZV and VZV in newborn animals.
**Cotton rat**	VZV replicates in cotton rat fibroblasts in vitro. VZV mutants can be tested.	Host proteins may not be conserved between cotton rats and humans, such that VZV proteins may not interact with cotton rat proteins the same way.
		Animals are aggressive.
		Limited species-specific immunologic reagents are available.
**SCID-hu mouse**	Differentiated human neurons and satellite cells, along with supportive tissues within the tissue microenvironment of the sensory ganglion, can be maintained long-term in vivo. Typical human neuronal subtypes are maintained, allowing analysis of VZV neurotropism at the single-cell level and comparison with HSV-1, the other neurotropic human alpha-herpesviruses. Viral determinants of VZV neurovirulence can be defined by evaluating VZV recombinants with targeted mutations to disrupt domains of VZV proteins. Opportunity to document the progression of VZV infection in sensory ganglia, demonstrating the capacity of VZV to induce fusion of neurons and satellite cells, unlike HSV-1. Opportunity to define innate defenses mounted against VZV infection by neural cells in vivo. Xenografts allow study of VZV infection of human DRG. Only available in vivo model for some human viruses. Better at reproducing the pathophysiology of human diseases and a better model for studying tissue tropism and antivirals. Relatively inexpensive compared to NHP. Allow monitoring and measuring viral replication in vivo using a bioluminescence assay.	Anterograde and retrograde transport of VZV between skin and ganglia cannot be modeled. Lacking human immune responses. Complicated surgery procedure. Large experimental variation. Human fetal tissues may be difficult to obtain. Absence of an adaptive immune system.
**NHP (SVV)**	Simian counterpart virus, SVV infects NHP, causes varicella, latent infection in ganglia and zoster that are similar to human VZV infection.	Expensive and need special approval. SVV reactivation results in a whole-body rash, while VZV reactivation in humans produces a dermatomal rash. SVV genome contains a unique ~3 kbp sequence at the left end, not present in VZV and potentially encoding species-specificity. Not the human virus, VZV. Pattern of infection is species-specific.

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
