# Peer review of "Current In Vivo Models of Varicella-Zoster Virus Neurotropism"

_viruses, 2019, doi:10.3390/v11060502_

Round 1
Reviewer 1 Report
The manuscript submitted by Mahalingam and colleagues provides a comprehensive review on the in vivo models currently being used to study VZV neurotropism. The review is authored by experts in the field of animal models to study VZV neuropathogenesis and is logical and well-written. I only have minor comments to improve the manuscript.
1. There are three figures included in the review. Given that SVV infection of non-human primates is such an important model to study neuropathogenesis and is a main focus of the review, I would suggest the authors include a figure showing some key features of the SVV non-human primate model (as they did for the VZV infection of SCID-hu ganglia models in Figs 1-2).
2. It would be useful to include for each model a brief discussion on whether the host immune response is intact and whether innate and or adaptive immunity can be studied in VZV neuropathogenesis. SVV is an excellent model where the role of both innate and adaptive host immune responses can be explored in this area.
3. Include references for the sentence on p10 line 11-12 “Such is also the case for VZV grown in vitro”
4. Include correct nomenclature for the VZV latency transcript (VLT) on page 11 line 11.
Author Response
The review is logical and well-written. We appreciate the comment.
1. Given that SVV infection of non-human primates is such an important model to study neuropathogenesis I would suggest the authors include a figure showing some key features of the SVV non-human primate model.
We have included a figure describing varicella and zoster in NHP. Page 14
2. It would be useful to include for each model a brief discussion on whether the host immune response is intact and whether innate and or adaptive immunity can be studied in VZV neuropathogenesis. SVV is an excellent model where the role of both innate and adaptive host immune responses can be explored in this area.
We have included a sentence to this effect in each of the models. Page 5 lines 19-21; Page 6 line 29; Page 9 lines 28-29; Page 12 line 28.
3. Include references for the sentence on p10 line 11-12 “Such is also the case for VZV grown in vitro”
We have included a reference in the revised manuscript. Weller, T.H. Serial propagation in vitro of agents producing inclusion bodies derived from varicella and herpes zoster. Proc. Soc. Exp. Biol. Med. 1953, 83, 340-346 Page 11, line17.
4. Include correct nomenclature for the VZV latency transcript (VLT) on page 11 line 11.
The nomenclature for VZV latency associated transcript has been corrected in the revised manuscript. page 4 line 31./
Reviewer 2 Report
This review concerns models of varicella zoster virus neurotropism. It is jointly co-authored by many of the leaders in the field. It discusses three different animal model systems that have been exploited to study VZV interactions with neurons in the establishment of neuronal infection and the latent state (the guinea pig, the cotton rat, the SCID-hu-DRG model) and also addresses the parallel model of SVV in non-human primates, which is acceptable, since SVV is closely related to VZV
1. While the review is fairly comprehensive, it reads as a disjointed and piecemeal work, which is an unlinked assembly from different authors of each model. The result reads rather as independently written and disjointed reviews of each system rolled into one, with parts repeated, sometimes more than twice. This does not read well. It is strongly suggested that the review should be rewritten and considered with some more oversight by one person (the lead author perhaps) in making it a cohesive review rather than individual contributions. It would also help to detail the advantages and disadvantages of each model in an honest manner that considers each models potential for contributing to future studies of VZV persistence and more importantly reactivation.
2. The abstract does not mention the guinea pig and cotton rat sections at all
3. The sections are quite different in their scope and depth. For example, the section on cotton rats is one page and quite succinct of all the work done, while the section on the SVV model is five pages and details almost everything done with the model, including many aspects that do not have that much relevance to neurotropism. Given the SVV model was also only just recently extensively reviewed in relation to its similarities for VZV in 2019 by Messaoudi, this SVV section should be considerably shortened. The same occurs for the guinea pig model:- it really is not necessary to go through all the old data regarding the past work on VZV in guinea pig culture, the early studies on transmission and the development of a faux rash in weanling hairless animals.
4. A minor comment is that the title does not accurately reflect SVV content, and that the SVV model should be taken with caution. It should also be acknowledged that SVV is a different virus, behaves differently from VZV in different non-human primate species, and has several differences to human VZV that should be more strongly emphasized. This is like having a title of “models of HSV-1 neurotropism” and then talking about PRV or BHV-1 latency. They are guides for VZV and what might happen, but do not tell us much of the human VZV and its features of neurotropism. They may hold true for both viruses, but then again, they may not. Some caution is warranted in the commentary
5. There are two figures concerning the SCID hu mouse model (each with a little overlap in how the model is set up in both the figure and the text for example) and only one diagrammatic representing the SVV model and its genome end comparison to VZV. It seems slanted and might be better to be more evenly distributed over the different models to make the review more cohesive.
SPECIFIC COMMENTS
6. Page 2 delete 9-18 and shorten 19-28, the data is not particularly helpful and is certainly not new
7. Page 2 line 47 stated animals did not become ill, then pg3 line 5 several signs of illness are detailed. Hypothermia, weight loss and rash in some This is confusing. In the original work they reported that VZV infection did cause signs of illness
8. Lines 14 to 18 do not make any sense. “In subsequent guinea pig studies….aimed at identifying immunological events….. and vaccine candidates..investigators found they could learn a great deal more from studying the SCID-hu model?” Huh? Particularly since the latter has no adaptive immune system, its useless to study vaccine immunity and the proteins that are immunogenic in SCID mice!!!
9. Pg 3 line 30-31 repeats human studies that are then again stated in lines 42-43. Line 30-31 are worded badly as it seems to be talking about the guinea pig and is actually talking about human studies in mid-sentence.
10. Line 41-43 stating that VZV is latent in ENS of almost every individual who has had the vaccine or varicella is over-concluding the work in the papers that are detailed. It is far from “in fact” this continues on page 4 line 2, quoting reference 39. There is no published data to suggest that virtually every neuron in the ENS and the DRG has a latent VZV infection. Only one ENS image is detailed in the paper the authors reference and there are no statistical analyses of other sections that qualify this statement. THE DATA IS ALSO REPORTED AS NOT SHOWN IN THAT PAPER. Furthermore the data regarding induced experimental reactivation of VZV by tacrolimus and a zoster like rash in reference 40 is also NOT SHOWN. Finally, other areas of this review point out that VZV latency is not associated with protein expression (page 6 line 48). This makes the guinea pig model confusing, since much of the work in the guinea pig model of persistence is based on the apparent expression of the ORF63 protein during latency. Most now agree that ORF63 protein is not made during human latency (Depledge et al 2018) so this is quite possibly a different type of event than is going on in human during latency.
11 in the cotton rat, it should be pointed out that it has never been established that VZV actually is able to replicate in the cotton rat in vivo. Furthermore, the evidence that infection of cotton rat fibroblasts results in a productive infection” is “DATA NOT SHOWN in the papers I looked up for this including the first works detailing ORF2deleti
on. This is rather an important point. A lack of replication, as seen in other species of the rat, means that latency cannot ever be reactivated- the persistent state seen in this model could be an abortive type of infection blocked at a mid stage of infection than latency. If this reviewer is wrong, please provide the documentation that VZV can replicate in cotton rats. This is an important issue.
Would this reviewer be correct in concluding that in all VZV mutants tested, if virus is produced in the inoculum then viral DNA reaches the ganglia. If virus is not produced in the inoculum, then virus does not reach the ganglia? This indicates that VZV DNA in the ganglia does not require any viral replication in the rat, yes? I am just trying to understand the usefulness of the cotton rat model and its limitations, and it may be that line 36, stating the cotton rat usefulness, should be extended to address these issues and acknowledge the limitations of replication clearly
Author Response
1. The review should be rewritten and considered with some more oversight by one person (the lead author perhaps) in making it a cohesive review rather than individual contributions.
We have made revisions in the manuscript so that it is more cohesive. We have also included a table addressing the advantages and disadvantages of each model. Page 17
2. The abstract does not mention the guinea pig and cotton rat sections at all. Guinea pig and cotton rat models are mentioned in the revised abstract.
3. SVV section and the guinea pig model should be considerably shortened.
The revised section on the guinea pig and SVV model are substantially shortened. (Pages 2-5). We believe that the early background studies using the guinea pig model is important because it sets the stage for the more modern work on this model. We have also removed a great deal of the detail, especially in the section on how VZV was attenuated to produce the vaccine. Please also see our response to the SPECIFIC COMMENT 6 of the reviewer.
4. A minor comment is that the title does not accurately reflect SVV content and SVV model should be taken with caution. It should also be acknowledged that SVV is a different virus, behaves differently from VZV in different non-human primate species, and has several differences to human VZV that should be more strongly emphasized. Some caution is warranted in the commentary.
We prefer to keep the title the same since SVV infection of NHP has served as a good animal model for VZV infection in humans. We do agree with the reviewer that SVV is a different virus and we have included this information in the manuscript.
5. There are two figures concerning the SCID hu mouse model (each with a little overlap in how the model is set up in both the figure and the text for example) and only one diagrammatic representing the SVV model and its genome end comparison to VZV. It seems slanted and might be better to be more evenly distributed over the different models to make the review more cohesive.
We have included an additional figure describing primary varicella as well as reactivation NHP (Figure 5).
SPECIFIC COMMENTS
6. Page 2 delete 9-18 and shorten 19-28, the data is not particularly helpful and is certainly not new.
The material that the referee characterized as “not particularly helpful” and “not new” has been greatly shortened. The original two paragraphs have been cut to one. We think, however, that it is necessary to cover the background information to provide insight into what led to the trial of guinea pigs as a model of the study of VZV. We agree that not all of the detail is required; however, the salient points provide a necessary frame of reference. Material used to introduce a subject in a review, is usually not new.
7. Page 2 line 47 stated animals did not become ill, then pg3 line 5 several signs of illness are detailed. Hypothermia, weight loss and rash in some is confusing. In the original work they reported that VZV infection did cause signs of illness.
We apologize for the lack of clarity. We did not say that the animals “did not become ill”. We said that they did not become visibly ill despite the mild rash hypothermia, and lack of weight gain, each of which was noted prior to the comment on the degree to which they appeared to be ill. Our point was that the guinea pigs were not obviously sick animals. We have changed the word visible to otherwise, to avoid the apparent contradiction. P. 3 line 5 does not detail signs of illness.
8. Lines 14 to 18 do not make any sense. “In subsequent guinea pig studies….aimed at identifying immunological events….. and vaccine candidates..investigators found they could learn a great deal more from studying the SCID-hu model?” Particularly since the latter has no adaptive immune system, its useless to study vaccine immunity and the proteins that are immunogenic in SCID mice.
We did not mean to imply that immunity and vaccines could be studied in SCID mice. We just meant to say that the investigators abandoned work of the guinea pig and turned to the study of VZV pathogenesis, which they have done to great effect by investigating VZV infection of human tissue in SCID-hu mice. We have now revised the text. Page 3 lines 11-13.
9. Pg 3 line 30-31 repeats human studies that are then again stated in lines 42-43. Line 30-31 are worded badly as it seems to be talking about the guinea pig and is actually talking about human studies in mid-sentence.
We thank the reviewer for the comment and apologize for the confusion. The material has been changed to distinguish between studies in human and guinea pigs. Page 3 line 38-42.
10. Line 41-43 stating that VZV is latent in ENS of almost every individual who has had the vaccine or varicella is over-concluding the work in the papers that are detailed. It is far from “in fact” this continues on page 4 line 2, quoting reference 39. There is no published data to suggest that virtually every neuron in the ENS and the DRG has a latent VZV infection. Only one ENS image is detailed in the paper the authors reference and there are no statistical analyses of other sections that qualify this statement. THE DATA IS ALSO REPORTED AS NOT SHOWN IN THAT PAPER. Furthermore the data regarding induced experimental reactivation of VZV by tacrolimus and a zoster like rash in reference 40 is also NOT SHOWN. Finally, other areas of this review point out that VZV latency is not associated with protein expression (page 6 line 48). This makes the guinea pig model confusing, since much of the work in the guinea pig model of persistence is based on the apparent expression of the ORF63 protein during latency. Most now agree that ORF63 protein is not made during human latency (Depledge et al 2018) so this is quite possibly a different type of event than is going on in human during latency.
The reviewer is correct that our original comment that “VZV is latent in ENS of almost every individual who has had the vaccine or varicella” was an overstatement. We have investigated the bowel of only a subset of individuals. Our statement is based on the observation that we have detected DNA and transcripts encoding VZV gene products in every fresh specimen of human gut that we have examined except that of controls, who were less than one year of age and lacked either vaccination or exposure to WT VZV. The references to our observations on the human ENS are numbers, 33, 34, 35, 36, and 37. We did not cite references 39 or 40 to support our discussion of the human ENS as the reviewer notes. References 39 and 40 describe infection of the guinea pig ENS. It is true that we have detected expression of ORF63 protein in guinea pig enteric neurons. Others have detected ORF63 protein in latently infected human neurons (Kennedy, Grinfeld, and Bell 2000 J. Virol 74, 11893-11898; Lungu et al. 1998. PNAS, 95: 7080-7085; Mahalingam et al. 1996; PNAS 93, 2122-2124). Zerboni et al find ORF63 protein in only a few neurons and rare ganglia. A good description of the data on ORF63 protein is found in reviews by Kennedy and Cohrs, 2010; J. Neurovirol. 16, 411-418 and Kennedy et al. 2015, J. Gen Virol, 96, 1581-1602). The general conclusion is that the expression of ORF63 protein in human ganglia is “controversial”. Depledge et al. 2018 agree with many other authors and confirm that transcripts encoding ORF63 are expressed in latently infected neurons along with transcripts encoding VLT; moreover, the transcripts “retain the potential to be translated during latency”. Depledge et al. 2018 point out that immunohistochemistry may not reveal the presence of VLT and ORF63 proteins because of the low abundance of the respective transcripts. They suggest that further studies with higher sensitivity are needed. One should also point out that human material has mainly been studied in neurons retrieved from cadavers, which have been fixed at a variety of intervals after death. Proteins in low abundance may be difficult to detect in such tissue due to autolysis. In contrast, the guinea pig neurons are investigated in tissue fixed immediately upon death of the animals; moreover, sensitive immunocytochemical methods have been utilized. It is thus not yet clear “that ORF63 protein is not made during human latency” (as the referee states) and it is certainly not clear that it is not made during latency in human enteric neurons. Page 4 lines 27-36.
We have however modified our comments, as requested, to indicate that “studies of the human gut have suggested that VZV may be latent in the ENS of at least many individuals who have either experienced varicella or been vaccinated. We do not think we should discuss the issue of whether or not the expression of ORF63 during latency in guinea pig enteric neurons suggests that latency is different in guinea pig or human. We have acknowledged the possibility. Page 4 lines 27-36.
11 In the cotton rat, it should be pointed out that it has never been established that VZV actually is able to replicate in the cotton rat in vivo. Furthermore, the evidence that infection of cotton rat fibroblasts results in a productive infection” is “DATA NOT SHOWN in the papers I looked up for this including the first works detailing ORF2deletion. This is rather an important point. A lack of replication, as seen in other species of the rat, means that latency cannot ever be reactivated- the persistent state seen in this model could be an abortive type of infection blocked at a mid-stage of infection than latency. If this reviewer is wrong, please provide the documentation that VZV can replicate in cotton rats. This is an important issue.
The reviewer is correct. Several years ago, we showed that VZV can infect cotton rat fibroblasts (resulting in cytopathic effects) and that virus could be passaged to uninfected cells by adding infected cells to uninfected cells. We did not publish the details of these experiments. Therefore, we have changed it to “unpublished observations”. Since VZV did not result in skin lesions or apparent organ disease in cotton rats after infection of the animals, it was unclear whether the virus could grow in other cell types and we did not pursue this further.
We have revised the first paragraph under the section on cotton rats as follows: It is important to note that neither acute infection (varicella) nor reactivation (zoster) have been shown to occur in cotton rats after inoculation with VZV. Thus, while VZV DNA and RNA are detected in ganglia of cotton rats which resembles the latent infection in human ganglia, the possibility of an abortive infection in the ganglia of cotton rats cannot be excluded. Page 5, lines 30-36.
Would this reviewer be correct in concluding that in all VZV mutants tested, if virus is produced in the inoculum then viral DNA reaches the ganglia. If virus is not produced in the inoculum, then virus does not reach the ganglia? This indicates that VZV DNA in the ganglia does not require any viral replication in the rat, yes? I am just trying to understand the usefulness of the cotton rat model and its limitations, and it may be that line 36, stating the cotton rat usefulness, should be extended to address these issues and acknowledge the limitations of replication.
As noted in the second paragraph of the section on cotton rats, VZV ORF21 is essential for the virus to replicate in cell culture, but it is not essential for the virus to establish latency. The mutant virus inoculum had to be produced in cell culture using complementing cells. Page 6 line 11.
Reviewer 3 Report
General Comments
1) This is an important review that puts together in one place all of the non tissue culture/cadaver models for study of infection of neurons by VZV. It is composed of several well-written separate “minireviews” by the experts in the respective models who for the most part, made the largest contributions to these studies in their own laboratories. To me, the “conclusion” section making some attempt at integrating information from the the various models is too brief, and really could use a table comparing and contrasting conclusions from the various models. Some words about which model(s) should be continued to be pursued to approach which outstanding issues would be valuable as well.
2) It would be extremely valuable to have a section comparing the 3 major experimental systems in which VZV neurotropism is studied, 1) cadaver/biopsy ganglia, 2) in-vitro systems using human stem-cell-derived neurons/neuroblastoma derived neuronal like cells, and 3) in-vivo models. Perhaps a table that would present the eminent authors of this review takes on the advantages/disadvantages/specific questions that can be asked or are inappropriate to ask with each model?
3) A major advance in the understanding of VZV latency, perhaps the most important and interesting aspect of its neurotropism, was published last year- the discovery of a potential latency-associated transcript VLAT by Depledge et al. The only minireviews that addresses it is a passing reference on pg 11 line 11 in the discussion of a previously discovered similar transcript in SVV. There should be at least one paragraph in the SVV minreview comparing and contrasting the human VLAT with the similarly located transcript in SVV, and potential implications/future directions that could be pursued specifically in the simian model for understanding its function. In addition, each of the other minireviews should acknowledge this exciting finding, and relate it to the model being reviewed.
Specific comments for each minireview
a) Guinea pigs
i) General comment. It has long seemed to me to be extremely odd that in spite of the long-known ability of VZV to infect GP cells, that was taken advantage of for generating the VZV vaccines, there has been almost no research using this model in the past few decades. Almost every paper on VZV contains in its introduction a sentence about the human specificity of the virus and the lack of small animal models. Indeed, with the exception of the Gershons, there seem to be no other groups that have used this model in the past few decades. Research on non-human primates is much more expensive and requires jumping through many more hoops to obtain ethical permissions, and yet there there are multiple groups using these animals to study SVV, which is not even the human virus!
Therefore, I think it is very important that the Gershons speculate why other groups are not using this model. It would be valuable for the readers of the review that are interested in pursuing VZV research if they could detail the experimental advantages/difficulties involved with the GP model.
In addition, most of the conclusion part of this section is devoted to ENS infection by VZV in the GP model. Although this has been the focus of work in the GP system, it seems to me that more could be said about how the hairless GP model is similar/dissimilar to human varicella/zoster in terms of the skin infection and non-enteric PNS infection.
ii) Pg 2, line16. In think that “several” decades is more appropriate than “many” decades.
iii) Pg 3 lines 37-42 The evidence for bona-fide reactivation of VZV in the guinea pig has only been provided from one experimental group, that of the Gershons. The only two papers in which this has been published are in supplements. Until these results are repeated by others (as pointed out by the Gershons themselves), it seems to me that the conclusions about latent/lytic infection, ORF61 and cell-free vs cell associated virus, are a bit strong for the amount of data available and this paragraphed should be written with a little less definitiveness
iv) Pg 4 first paragraph This paragraph is based on a lecture, not results published in a peer-reviewed journal. Obtaining reactivation of VZV in the skin is a striking and important result, and should be documented appropriately. Therefore, I think that the results mentioned in this paragraph should be supported by figures/tables, as were the results in the SCID-hu model for the neurotropic aspect of VZVORF7. It is not appropriate in my opinion to include such detailed results/conclusions without publishing the data here or in a conventional format in a peer-reviewed forum.
v) Pg 4 line 10 The authors report detection of VZV DNA and transcripts, did they obtain infectious virus?
b) Cotton rats
i) General Comment 1: In many papers using the CR model the expression “latent infection” is used. The presence of VZV DNA and some transcripts is the experimental evidence used in these studies for latent infection of neurons. But, latency, whether at the organismal or cellular level, is the maintenance of reactivatable genomes. Without demonstrating that VZV can be reactivated in infected CR, these are likely abortive and not latent infections. Such infections have been studied in-vitro in non-human CHO cells for example (Finnen et al, 2006). This therefore raises the question of the ability of studies using CR to advance our understanding of VZV latency. Perhaps in the conclusion it could be suggested that cotton rat cells/neurons could be used to study restriction of VZV replication, as opposed to latent infection
ii) General comment 2) The last publication mentioned in the review was in 2010. Therefore, this model has not been used for over a decade. This section should therefore explain why a) the model was only used by one group when it was used and b) why it has ceased to be used by any research groups.
iii) The section discusses the involvement of multiple VZV genes in latency, and cites the presence of these transcripts in cadaver ganglia. Recent studies of post-mortem human ganglia removed soon after death showing almost no expression of VZV transcripts, and a more recent paper describing the putative VZV latency-associated transcript VLAT, have demonstrated that it is likely that only VLAT and ORF63 are consistently associated with latency in human ganglia. The text should reflect these new findings, and the implications for the cotton rat studies.
iv) Pg 5 line 24. In Ambagala et al, I did not see experimental evidence that the engineered viruses were “”impaired for virus reactivation”, as stated in the text. The paper speaks about impairment of replication in culture, not reactivation. Therefore, this text should be modified to reflect the results obtained in the study.
c) SCID-hu
i) General comment Important and valuable information has been gleaned, and is continuing to be obtained from the SCID-hu model for VZV neurotropism. Infection is initially productive, and apparently switches to a latency-like state. Unfortunately, and surprisingly, the few groups that have used this system have not published reactivation of VZV from the xenografts. Considering that DRG xenografts can be maintained for months, it would be extremely interesting to use this model for studying, for example, the expression profile over time of reactivation.
ii) General comment 2: The SCID-hu model has the tremendous advantage over the small animal models in that human neurons in their normal 3D topography are are used. This has lead to the fascinating observations regarding neuronal subtype infection, satellite cell infection, and differences between HSV1 and VZV infection. However, it should be pointed out at some point that there are two complicating issues with this model. First, the transplants are ectopic, that is, under the kidney capsule, rather than orthotopic. Therefore, the DRG axons can not grow out to innervate their normal targets in the skin, where they may be infected in varicella disease. Transplantation of peripheral ganglia to ectopic positions, even of adult ganglia, has long know to elicit important changes in neurons, to the point of changing neurotransmitters (i.e. Shotzinger Dev. Neurobiol 25 620-639 1994, Stanke et al Development 2006 133: 141-150). Secondly, the ganglia are taken from human fetuses around mid gestation, and may not reflect the situation in adult ganglia. Important changes in the DRG have been documented in murine DRG at a parallel time of development, including massive cell death periods that sculpt the final distribution of physiological and molecular phenotypes in the ganglia.
iii) The authors, who continue to take advantage of this elegant experimental system to reveal new and interesting information, do not present any data on the presence/absence of the putative VLAT transcript in DRG explanted to mice. VLAT should be mentioned, if only to say that it has not yet been assayed for in this model.
iv) Pg 6 lines 17 – pg 8 line 2 This is all a single paragraph with around 35 lines. Perhaps the authors could break it into a few paragraphs to enhance readability?
v) Section 4.1 This section presents new, unpublished results. In my opinion, they are interesting and exciting enough to warrant publication in a conventional article and not a review (this is of course is the authors prerogative to decide, just my thoughts here). The legend to figure 2 should have a little more description, I assume the “growth curves” that are described by “photon counts” on the Y axis, are quantitative measurements made from mice as shown in C? the pictures in figure 2F are so tiny that its not possible to see whether signal is in neurons or glia. “not detected in any of the samples”, how many samples were tested?
d) SVV
i) Pg 10 line 28. Perhaps homology to VZV at the amino acid level rather than the nucleotide level would be more informative to the reader for getting an impression as to the conservation of the ORFs?
ii) There are 3 duplicated ORFs present in VZV, is this the case for SVV? Are there ORFs shown to be important in VZV that are missing in SVV? Or vice-versa? In a comparative review, this information would be valuable.
iii) Pg 11 lines 10-12 As mentioned above, the recent discovery of VLAT may change the way we think about VZV latency. The presence of a homologous transcript in SVV is therefore potentially of utmost importance. Please compare the length of the simian and human transcripts, splicing information for SVV if known, relative prevalence etc.
iv) Pg 11 line 26 I would emphasize that the NHP model is the only one where there is an initial infection resembling varicella in at least one type of monkey and infection route.
v) A table comparing the disease manifestations/infection routes etc of the primary NHP that have been studied would help those of us who don’t work with this model.
vi) Pg 12 line 51 All reviews of VZV neurotropism point out that infection of peripheral ganglion neurons may occur via the viremia or by retrograde infection from axonal endings in the skin. The strong evidence that VZV-infected lymphocytes are present in ganglia as little as 3 days after infection would seem to give strong preference to the hematogenous route of infection and this could be emphasized I believe.
vii) There is a bit too much experimental emphasis on the levels of immunosuppressive drugs given to the monkey in section 5.3. It makes the reading a bit dense, and hard to pick out the important results.
viii) Several aspects of VZV disease and study that are important, seem to have not been addressed in the SVV system. PHN, shedding in the saliva, giant cell arteritis and corneal reactivation disease are all topics of current interest which to me would best be studied in SV infection of NHP. Perhaps the authors of this section could address why these clinical topics have not been (or have they been?) addressed in the SVV model.
e) Conclusion
i) A table comparing and contrasting the models to varicella and zoster in humans in a table would be extremely useful for integrating this information. The table should compare at least the NHP SVV model to VZV diseases (and then could be in the previous section instead of the final conclusion).
ii) The conclusion about the cotton rat data is not warranted, please see comment b)i) above
iii) The reasons that the GP model is not more widely used should be mentioned here.
iv) Integration of the discovery of VLAT into the picture of VZV neurotropism should be included here, especially since there is a similar transcript expressed in SVV latency.
Author Response
General Comments
1) This is an important review that puts together in one place all of the non tissue culture/cadaver models for study of infection of neurons by VZV. It is composed of several well-written separate “mini reviews” by the experts in the respective models who for the most part, made the largest contributions to these studies in their own laboratories. To me, the “conclusion” section making some attempt at integrating information from the various models is too brief, and really could use a table comparing and contrasting conclusions from the various models. Some words about which model(s) should be continued to be pursued to approach which outstanding issues would be valuable as well.
We have now revised the conclusion section and included a Table at the end of the document. Pages 16 and 17.
2) It would be extremely valuable to have a section comparing the 3 major experimental systems in which VZV neurotropism is studied, 1) cadaver/biopsy ganglia, 2) in-vitro systems using human stem-cell-derived neurons/neuroblastoma derived neuronal like cells, and 3) in-vivo models. Perhaps a table that would present the eminent authors of this review takes on the advantages/disadvantages/specific questions that can be asked or are inappropriate to ask with each model?
We have compared the animal (in vivo) models described in this review in Table 1. We believe that in vitro systems using human stem-cell derived neurons is outside the scope of this review.
3) There should be one paragraph comparing and contrasting the human VLAT with the similarly located transcript in SVV, and potential implications/future directions that could be pursued specifically in the simian model for understanding its function. In addition, each of the other mini reviews should acknowledge this exciting finding, and relate it to the model being reviewed.
We have acknowledged VLT in each of the animal models. Page 4, lines27-36; Page5, line 35; Page 7, lines 45-47; Page 12 lines 18-20. We have also introduced the following statement in the revised manuscript: VLT is a spliced transcript which encodes a protein with late expression kinetics in lytically infected cells in vitro and in zoster lesions. The VLT suppresses ORF 61 gene expression in co-transfected cells suggesting a mechanism by which it may contribute to maintenance of viral latency. The simian varicella model provides a useful approach to further explore the role of VLT in viral latency. Page 12, line 26.
Specific comments for each mini review
a) Guinea pigs
i) General comment. It has long seemed to me to be extremely odd that in spite of the long-known ability of VZV to infect GP cells that was taken advantage of for generating the VZV vaccines there has been almost no research using this model in the past few decades. Almost every paper on VZV contains in its introduction a sentence about the human specificity of the virus and the lack of small animal models. Indeed, with the exception of the Gershons, there seem to be no other groups that have used this model in the past few decades.
Research on non-human primates is much more expensive and requires jumping through many more hoops to obtain ethical permissions, and yet there are multiple groups using these animals to study SVV, which is not even the human virus!
We agree with the reviewers about the difficulty in working with the NHP. We have included this information in the revised manuscript in Table1.
Gershons should speculate why other groups are not using the guinea pig model. It would be valuable for the readers of the review that are interested in pursuing VZV research if they could detail the experimental advantages/difficulties involved with the GP model. In addition, most of the conclusion part of this section is devoted to ENS infection by VZV in the GP model. Although this has been the focus of work in the GP system, it seems to me that more could be said about how the hairless GP model is similar/dissimilar to human varicella/zoster in terms of the skin infection and non-enteric PNS infection.
This is indeed a difficult comment to address. The guinea pig model, used by the Gershons, is a valuable one for all of the reasons indicated by the reviewer. One can only speculate about what might motivate other investigators; however, it is possible that early observations of Myers et al. (1991) which showed that varicella symptoms were not mimicked in guinea pigs discouraged scientists in search of an animal model. At the time these studies were done, an immune response, symptoms of a disease, and culturing of virus were the primary tools used to investigate VZV. There was no good method to detect latent VZV. The introduction of molecular biological techniques were very helpful. Investigators have been, and are still, very aware of Koch’s postulates, one of which is to demonstrate that injection of an infectious agent into animals mimics a human disease. The efforts of Myers et al. confirmed that VZV infects guinea pigs, but they clearly failed in their primary purpose, which was to mimic varicella in animals. This point is well illustrated in comment #3 of Reviewer 2: “it really is not necessary to go through all the old data regarding the past work on VZV in guinea pig culture, the early studies on transmission and the development of a faux rash in weanling hairless animals.” To this day, the early work on guinea pigs is still undervalued. Arvin et al., made good use of the ability of VZV to infect guinea pigs to study immunity to the agent while the Gershons have pursued latency and reactivation. Page 3, lines11-13.
Investigations of guinea pigs are difficult since the animals breed slowly and are very expensive. Guinea pig husbandry and investigation are also more strictly regulated than mice or rats.
It is difficult in the absence of a survey, to speculate about why other groups have not used this model. The review outlines why the Gershons were motivated to select it. We have amplified our discussion of early experiments that revealed the failure of varicella to be mimicked in guinea pigs. There has, however, been utilization of the guinea pig model, particularly for the work on isolated enteric neurons, by other groups. They include: Walters et al., J Virol, 2008. 82(17): p. 8673-86 and Stallings et al., J Virol, 2006. 80(3): p. 1497-512 and Ambagala, et al., J Virol, 2009. 83(1): p. 200-9. These publications have not been included in the review because they are specialized and not within the focus of this review.
ii) Pg 2, line16. In think that “several” decades is more appropriate than “many” decades. This sentence was deleted as response to reviewer 2.
iii) Pg 3 lines 37-42 The evidence for bona-fide reactivation of VZV in the guinea pig has only been provided from one experimental group. The only two papers in which this has been published are in supplements. Until these results are repeated by others, it seems to me that the conclusions about latent/lytic infection, ORF61 and cell-free vs cell associated virus, are a bit strong for the amount of data available and this paragraphed should be written with a little less definitiveness.
The revised manuscript has been modified to be less definitive. Now we state that the infection of neurons appeared to be lytic. We also state that cell-free virus led to apparent virus latency. To improve clarity, we also specified exactly what the relevant observations actually were so that readers could understand what was meant when infection was described as lytic (expression of VZV glycoproteins; rapid cell death), latent (no glycoprotein expression, survival for extended periods of time), or reactivating (expression of VZV glycoproteins, electron microscopic detection of gE-immunoreactive virions, transmission of infection to co-cultured MeWo cells, and death of infected neurons within 48-72 hours). It is implied that reactivation of VZV in neurons with apparent latency is a confirmation of the idea that the infection before reactivation was latent. The best evidence for latency is reactivation. Page5
iv) Pg 4 first paragraph is based on a lecture and not published in a peer-reviewed journal. Obtaining reactivation of VZV in the skin is a striking and important result, and should be documented appropriately. Therefore, I think that the results mentioned in this paragraph should be supported by figures/tables, as were the results in the SCID-hu model for the neurotropic aspect of VZVORF7. It is not appropriate in my opinion to include such detailed results/conclusions without publishing the data here or in a conventional format in a peer-reviewed forum.
Although the paragraph describes data that were presented in a lecture, the accompanying manuscript was indeed peer reviewed. The editors who sent the paper out for review are outstanding investigators of herpes viruses. It is because the observations were peer reviewed that they cannot be published elsewhere in what might be considered “a conventional format”. We have, however, added a figure that illustrates the patch of green fluorescence on the skin of a guinea pig that follows reactivation with tacrolimus and corticotrophin-releasing hormone. Page 4, Figure 1.
v) Pg 4 line 10 The authors report detection of VZV DNA and transcripts, did they obtain infectious virus?
Infectious virus was recovered from reactivating guinea pigs. Those data are in a manuscript that is in preparation and thus we prefer not to present them in this review.
b) Cotton rats
i) General Comment 1: In many papers using the CR model the expression “latent infection” is used. The presence of VZV DNA and some transcripts is the experimental evidence used in these studies for latent infection of neurons. But, latency, whether at the organismal or cellular level, is the maintenance of reactivatable genomes. Without demonstrating that VZV can be reactivated in infected CR, these are likely abortive and not latent infections. Such infections have been studied in-vitro in non-human CHO cells for example (Finnen et al, 2006). This therefore raises the question of the ability of studies using CR to advance our understanding of VZV latency. Perhaps in the conclusion it could be suggested that cotton rat cells/neurons could be used to study restriction of VZV replication, as opposed to latent infection.
As noted in the response to reviewer 2 above, we have now added sentences to the first paragraph in the cotton rat section, indicating that while the cotton rat has been used as a model for latency (since viral nucleic acids in ganglia resemble the human ganglia), an abortive infection cannot be excluded. We prefer not to speculate about the use of cotton rat neurons to study restriction of replication, but instead as we have done to indicate the limitations of the model. Page 5, lines 30-36; Page 6, lines26-29.
ii) General comment 2) The last publication mentioned in the review was in 2010. Therefore, this model has not been used for over a decade. This section should therefore explain why a) the model was only used by one group when it was used and b) why it has ceased to be used by any research groups.
We have added the following information to the end of the section on cotton rats: In addition, compared with other small animal models, cotton rats are difficult to work with since they are very aggressive animals, immunologic reagents are much more limited for these animals than for mice, and fewer suppliers sell these animals; accordingly, this model has not been used for studies of VZV by researchers other than those who originally reported its use. Page 6, lines 25-29.
iii) The section discusses the involvement of multiple VZV genes in latency, and cites the presence of these transcripts in cadaver ganglia. Recent studies of post-mortem human ganglia removed soon after death showing almost no expression of VZV transcripts, and a more recent paper describing the putative VZV latency-associated transcript VLAT, have demonstrated that it is likely that only VLAT and ORF63 are consistently associated with latency in human ganglia. The text should reflect these new findings, and the implications for the cotton rat studies.
The section on cotton rats begins (first paragraph) and ends (last paragraph) with statements that a transcript expressed during latency in human ganglia (ORF63) is also expressed in cotton rat ganglia, while a transcript rarely expressed during latency in humans (ORF40) is also usually not expressed in latently infected cotton rats. Thus, this is the same pattern of gene expression as in the VLAT paper (only VLAT and ORF63 are consistently expressed). While mutants in multiple VZV genes are discussed in this section, we have not reported expression of VZV transcripts from these multiple genes in the cotton rat. Page 5, line35-36.
iv) Pg 5 line 24. In Ambagala et al, I did not see experimental evidence that the engineered viruses were “impaired for virus reactivation”, as stated in the text. The paper speaks about impairment of replication in culture, not reactivation. Therefore, this text should be modified to reflect the results obtained in the study.
The reviewer is correct, and we thank the reviewer for pointing this out. We have changed the sentence in the fourth paragraph in the section on cotton rats to: The cotton rat model was used for development of a VZV mutant that was impaired for latency, but not for replication in vitro. Page 6, line 11.
c) SCID-hu
i) General comment Important and valuable information has been gleaned, and is continuing to be obtained from the SCID-hu model for VZV neurotropism. Infection is initially productive, and apparently switches to a latency-like state. Unfortunately, and surprisingly, the few groups that have used this system have not published reactivation of VZV from the xenografts. Considering that DRG xenografts can be maintained for months, it would be extremely interesting to use this model for studying, for example, the expression profile over time of reactivation.
Investigation of VZV reactivation in the SCID-hu DRG model has been challenging due to absence of a marker to identify latently infected neurons. Recently, Depledge and colleagues (2018) identified a spliced VZV mRNA antisense to VZV open reading frame 61 (VLT) that is expressed during latency in human neurons. Page 4, lines 30-37.
ii) General comment 2: The SCID-hu model has the tremendous advantage over the small animal models in that human neurons in their normal 3D topography are used. This has lead to the fascinating observations regarding neuronal subtype infection, satellite cell infection, and differences between HSV1 and VZV infection. However, it should be pointed out at some point that there are two complicating issues with this model. First, the transplants are ectopic, that is, under the kidney capsule, rather than orthotopic. Therefore, the DRG axons cannot grow out to innervate their normal targets in the skin, where they may be infected in varicella disease. Transplantation of peripheral ganglia to ectopic positions, even of adult ganglia, has long known to elicit important changes in neurons, to the point of changing neurotransmitters (i.e. Shotzinger Dev. Neurobiol 25 620-639 1994, Stanke et al Development 2006 133: 141-150). Secondly, the ganglia are taken from human fetuses around mid-gestation, and may not reflect the situation in adult ganglia. Important changes in the DRG have been documented in murine DRG at a parallel time of development, including massive cell death periods that sculpt the final distribution of physiological and molecular phenotypes in the ganglia.
As we had mentioned in our manuscript xenografts were not used until 20 weeks after implantation because by this time point, we were able to show a high degree of differentiation using neural markers. At 20 weeks, nociceptive and mechanoreceptive neurons, the major neuronal subtypes present in postnatal and adult human ganglia, were readily distinguished using peripherin immunoreactivity and RT97 immunoreactivity, respectively. Page 7 line 3-4 .
iii) The authors, who continue to take advantage of this elegant experimental system to reveal new and interesting information, do not present any data on the presence/absence of the putative VLAT transcript in DRG explanted to mice. VLAT should be mentioned, if only to say that it has not yet been assayed for in this model.
In the revised manuscript we have added the statement: The presence of VLT has not been assayed in the SCID-hu model. Page7, line 48-49.
iv) Pg 6 lines 17 – pg 8 line 2 This is all a single paragraph with around 35 lines. Perhaps the authors could break it into a few paragraphs to enhance readability?
We have introduced a paragraph on page 7 line
v) Section 4.1 This section presents new, unpublished results. In my opinion, they are interesting and exciting enough to warrant publication in a conventional article and not a review (this is of course is the authors prerogative to decide, just my thoughts here). The legend to figure 2 should have a little more description, I assume the “growth curves” that are described by “photon counts” on the Y axis, are quantitative measurements made from mice as shown in C? the pictures in figure 2F are so tiny that its not possible to see whether signal is in neurons or glia. “not detected in any of the samples”, how many samples were tested?
This section has been previously published (75, 76 and 77). The Fig. 2 legend has been revised. Fig. 2F has been published, and detailed results were described in (77).
d) SVV
i) Pg 10 line 28. Perhaps homology to VZV at the amino acid level rather than the nucleotide level would be more informative to the reader for getting an impression as to the conservation of the ORFs? The revised manuscript states,” SVV DNA encodes 69 distinct SVV ORFs, each sharing 27- 75% amino acid identity….” Page 11, lines 36-37; Page 12 line 1.
ii) There are 3 duplicated ORFs present in VZV, is this the case for SVV? Are there ORFs shown to be important in VZV that are missing in SVV? Or vice-versa? In a comparative review, this information would be valuable.
Like VZV DNA, the SVV genome includes three ORFs (62, 63, and 64) that are duplicated within the S segment. However VZV ORF2 is missing in the SVV genome. The other differences between the two genomes are shown in Figure 4. Page 11, lines 36-37; Page 12 lines1-3
iii) Pg 11 lines 10-12 As mentioned above, the recent discovery of VLAT may change the way we think about VZV latency. The presence of a homologous transcript in SVV is therefore potentially of utmost importance. Please compare the length of the simian and human transcripts, splicing information for SVV if known, relative prevalence etc.
Although SVV LAT was discovered first it is not as well characterized as VLT. Page 12, lines 25-26.
iv) Pg 11 line 26 I would emphasize that the NHP model is the only one where there is an initial infection resembling varicella in at least one type of monkey and infection route.
The following sentence has been added to the revised manuscript: SVV infection of nonhuman primates provides the only experimental animal model that closely simulates human varicella. Page 12, lines 27-28.
v) A table comparing the disease manifestations/infection routes etc of the primary NHP that have been studied would help those of us who don’t work with this model.
We have revised the section on NHP to clarify the models and results. Page 13 lines 4-8.
vi) Pg 12 line 51 All reviews of VZV neurotropism point out that infection of peripheral ganglion neurons may occur via the viremia or by retrograde infection from axonal endings in the skin. The strong evidence that VZV-infected lymphocytes are present in ganglia as little as 3 days after infection would seem to give strong preference to the hematogenous route of infection and this could be emphasized I believe.
We have clarified the hematogenous route of infection in the revised manuscript Page13 lines 37-38.
vii) There is a bit too much experimental emphasis on the levels of immunosuppressive drugs given to the monkey in section 5.3. It makes the reading a bit dense, and hard to pick out the important results.
We have shortened the details in the revised manuscript. Pages 12-15.
viii) Several aspects of VZV disease and study that are important, seem to have not been addressed in the SVV system. PHN, shedding in the saliva, giant cell arteritis and corneal reactivation disease are all topics of current interest which to me would best be studied in SV infection of NHP. Perhaps the authors of this section could address why these clinical topics have not been (or have they been?) addressed in the SVV model.
Recently, we have shown that SVV DNA in saliva can be used to monitor acute infection in RM [153] Future studies using the NHP model will include these aspects. Page 16 lines 12-15.
e) Conclusion
i) A table comparing and contrasting the models to varicella and zoster in humans in a table would be extremely useful for integrating this information. The table should compare at least the NHP SVV model to VZV diseases (and then could be in the previous section instead of the final conclusion).
Please see Table1 at the end of the document.
ii) The conclusion about the cotton rat data is not warranted, please see comment b) i) above.
We stated that “The cotton rat model has been used not only to identify the virus genes required for ganglionic infection, but also to show restricted transcription of VZV genes in ganglia weeks after experimental inoculation.” We only claim that with the use of viral mutants, we have shown that certain viral genes are required ganglionic infection (i.e. for virus to be detected in ganglia), not whether the infection can be associated with full virus replication or reactivation. In addition, as noted in the response to reviewer 3, comment b iii we find the same restricted transcription of VZV genes (IE63 only) as has been seen in humans. Thus, we think the statement in the conclusions about the cotton rat model is accurate.
iii) The reasons that the GP model is not more widely used should be mentioned here.
We have added the limitation that guinea pigs do not completely mimic human VZV infection in that primary varicella is not seen. Table 1
iv) Integration of the discovery of VLAT into the picture of VZV neurotropism should be included here, especially since there is a similar transcript expressed in SVV latency.
We have incorporated the information about VLT into the section on SVV. Page12, lines 25-26.
Reviewer 4 Report
Its clear multiple animal model systems are required to investigate the various aspects of virus-host interactions of Varicella-zoster virus (VZV). This review contributed by some of the key labs in the field describe the guinea pig, cotton rat, and SCID-hu mouse models of VZV as well as SVV infection of non-human primates. Overall, this methodical review is a significant contribution to the VZV field and the general virology community.
Minor comments:
-Surprising that work in the lab of Paul Kinchington and earlier studies by Fleetwood Walker on the VZV model of post-herpetic neuralgia was not included in this review.
Author Response
Its clear multiple animal model systems are required to investigate the various aspects of virus-host interactions of Varicella-zoster virus (VZV). This review contributed by some of the key labs in the field describe the guinea pig, cotton rat, and SCID-hu mouse models of VZV as well as SVV infection of non-human primates. Overall, this methodical review is a significant contribution to the VZV field and the general virology community.
We appreciate the reviewer’s comment.
Minor comments:
-Surprising that work in the lab of Paul Kinchington and earlier studies by Fleetwood Walker on the VZV model of post-herpetic neuralgia was not included in this review.
We apologize for the unintentional exclusion. The rat model described by Drs. Kinchington and Walker is indeed valuable to VZV-induced neurological diseases. We have included a reference [154] to this work in the conclusion section.
Round 2
Reviewer 2 Report
This resubmitted article is a review of the animal models for VZV and neurotropism. By and large it is massively improved. it comes across as not an amalgam of different authors, but rather a concerted work. it is well thought out.
i think that it is now acceptable for